# Lagrangian Meets Diffusion: Feasibility-aware Generative Modeling for Mixed Integer Linear Programming

## Abstract

End-to-end Predict-and-Search (PaS) methods show promise for Mixed Integer Linear Programming (MILP), but they typically assume variables independence and provide only deterministic single-point predictions, limiting solution diversity and demanding extensive search for high-quality solutions. We propose **VRG**, a feasibility-aware generative framework that operates in visual space. It transforms MILP solution vectors into image representations, which are in turn processed by a U-Net-based score network with Lagrangian relaxation guidance. The visual encoding enables convolutional kernels to capture interdependencies among variables while Lagrangian relaxation guides sampling toward feasible, near-optimal regions. The guided generator produces diverse, high-quality candidates rather than a single point estimate. The resulting candidates define compact and effective trust-region subproblems for standard MILP solvers. Across various public benchmarks, VRG consistently outperforms PaS baselines in solution quality and, while maintaining competitive optimality with state-of-the-art solvers such as SCIP and Gurobi, achieves markedly lower computational effort (reduced search time and explored nodes). Our source code is available at https://anonymous.4open.science/r/VRG-E09E/.

## 1 Introduction

Mixed Integer Linear Programming (MILP) models a wide range of finite combinatorial optimization problems, a canonical class of NP-hard challenges (Karp, 1972a). MILP has been extensively applied in production planning (Ye et al., 2023), resource allocation, and energy management (Pinzon et al., 2017), as well as in emerging applications across advanced manufacturing and industry. Owing to their intrinsic computational complexity, solving MILP instances in polynomial time is generally intractable. Classical techniques, such as Branch and Bound (Land & Doig, 1960a) and Cutting-Plane methods (Gomory, 1958a), remain the foundation for attaining optimality, but they often incur high computational costs on large-scale instances, limiting their practicality in time-sensitive real-world applications.

Recently, machine learning has shown great potential in accelerating MILP solving by leveraging structural similarities across problem instances and reusing decision knowledge (Gasse et al., 2019; Han et al., 2023; Gupta et al., 2020; Ling et al., 2024). In this paper, we focus on one specifically promising direction (Han et al., 2023), known as Predict-and-Search (PaS). PaS methods utilize GNN to predict the marginal probability of each variable, identifying promising regions or critical components of the solution space where high-quality solutions are likely to reside, and then apply local search algorithms within the reduced problem to find optimal solutions.

However, existing PaS methods generally encounter the following limitations:

i. PaS models typically assume that decision variables across different dimensions of the solution space are independent. This assumption deviates from the actual structure of MILP problems, where variables are often tightly coupled due to shared constraints & objectives.

ii. The approximate solutions predicted by PaS models are not guaranteed to be feasible or optimal, which imposes an inherent upper bound on the quality of final solutions.

Figure 1: Relaxation guided solution generation illustration process. Solutions sampled from the learned distribution progressively enter the linear feasible region while approaching the optimal solution.

iii. Predictions are often binarized using deterministic ranking strategies (e.g., based on thresholds $k_0$ and $k_1$), resulting in singular and rigid solution structures. This over-simplification can propagate to the construction of trust regions, reducing robustness and limiting exploration of the feasible space.

To address these limitations, we propose a feasibility-aware generative framework for MILP that performs visual relaxation score generation. Our approach transforms solution vectors into visual representations, enabling convolutional neural networks to capture interdependencies among decision variables via convolutional receptive fields.

In parallel, we construct a Lagrangian-guided probability distribution and derive the corresponding reverse-time stochastic differential equation (SDE)(Song et al., 2021; Karras et al., 2024), creating a score-based generative model with theoretical guarantees on both feasibility and optimality. Starting from Gaussian noise, this model progressively denoises toward high-quality feasible regions Fig.1). The resulting guided distribution remains multimodal, facilitating diverse sampling while improving both feasibility and optimality.

These diverse, guided samples enable the construction of more robust, higher-quality subproblems within compact trust regions, unlike conventional PaS methods that rely on singular, hard-binarized solutions. As a result, our method improves search robustness and efficiency and yields higher-quality solutions for MILPs. Our key contributions are as follows:

- We propose a feasibility-aware generative framework that integrates Lagrangian relaxation with diffusion to learn a guided generator that explicitly steers sampling toward feasible, high-quality regions. Our analysis establishes an optimization-equivalence result and a concentration guarantee that tightens the effective search space, providing formal guarantees on both feasibility and near-optimality and addressing the quality limitations of PaS approaches.

- We transform solution vectors into visual image representations, enabling convolutional neural networks to capture variable interdependencies through spatial receptive fields, overcoming the variable independence assumption in PaS.

- We generate a multimodal distribution of high-quality solutions from Gaussian noise, avoiding PaS's singular, hard-binarized solutions and improving robustness in downstream search.

## 2 RELATED WORK

MILP problems are among the most challenging combinatorial optimization tasks due to their hybrid decision spaces. Classical solving methods, such as branch-and-bound (Gomory, 1958b) and cutting-plane algorithms (Land & Doig, 1960b), address MILPs by systematically exploring, simplifying, or relaxing the original formulation. However, these approaches can exhibit exponential time complexity in the worst case, limiting scalability for large-scale or real-time applications. Notable ML-based directions include diffusion model based MILP solvers that exploit specific structural

properties of MILP instances (Sun & Yang, 2023; Yu et al., 2024; Sanokowski et al., 2024), Predict-and-Search (PaS) frameworks (Han et al., 2023; Huang et al., 2023), and hybrid approaches that integrate machine learning into the traditional branch-and-bound process (He et al., 2014; Gasse et al., 2019; Chmiela et al., 2021; Paulus & Krause, 2023). However, existing approaches suffer various limitations: PaS methods typically lack convergence guarantees and do not provide feasibility guidance during search, while diffusion-based solvers are predominantly constrained to problems with specific structural properties. There remains a critical need for PaS frameworks with theoretical convergence assurances and systematic, feasibility-aware guidance to accelerate solving and improving solution quality across diverse MILP instances.

## 3 PRELIMINARIES

**Mixed Integer Linear Programming (MILP)** seeks an optimal solution to a linear objective subject to both linear constraints and integrality restrictions. For an instance $s$ with $m$ constraints and $n$ variables, the standard MILP formulation is:

$$\min_{x \in \mathbb{R}^n} c^\top x \quad \text{s.t.} \quad Ax \le b, \, l \le x \le u, \, x_j \in \mathbb{Z} \, \forall j \in I \tag{1}$$

where $A \in \mathbb{R}^{m \times n}$, $c \in \mathbb{R}^n$, $b \in \mathbb{R}^m$, $l, u \in (\mathbb{R} \cup \{\pm\infty\})^n$, and $I \subseteq \{1, \dots, n\}$ denotes the set of indices for integer-constrained variables. The feasible region is:

$$D = \left\{ x \in \mathbb{R}^n \mid Ax \le b, \, l \le x \le u, \, x_j \in \mathbb{Z}, \, \forall j \in I \right\}.$$

**Lagrangian relaxation** Relaxation techniques for MILPs aim to simplify complex constraints, thus reducing computational overhead. Among these strategies, Lagrangian relaxation is particularly effective. For the MILP in Eq.(1) with optimal value $z_{\text{MILP}}^*$, its Lagrangian relaxation is constructed by dualizing the constraints $Ax \le b$ with Lagrange multipliers $\lambda \ge 0$:

$$z_{\text{Lagrangian}}(\tilde{x}, \lambda) = c^\top \tilde{x} + \lambda^\top (A\tilde{x} - b) \tag{2}$$

where $\tilde{x}$ denotes a relaxed solution. It is well-established that optimizing over $\lambda$ provides a valid lower bound on the MILP optimum $z_{\text{MILP}}^*$.

**Theorem 1** (Dual Convergence from (Shor, 1985))**.** *Consider the Lagrangian dual problem for a given trial solution $\tilde{x}$:*

$$z_{LD}(\tilde{x}) = \max_{\lambda \ge 0} z_{Lagrangian}(\tilde{x}, \lambda). \tag{3}$$

*Let $\{\lambda^{(k)}\}$ be the sequence of multipliers generated by the subgradient method with step sizes $\{\alpha_k\}$ satisfying:*

$$\alpha_k > 0, \quad \sum_{k=1}^{\infty} \alpha_k = \infty, \quad \sum_{k=1}^{\infty} \alpha_k^2 < \infty. \tag{4}$$

*Then the sequence of dual objective values converges to the optimal dual value:*

$$\lim_{k \to \infty} z_{Lagrangian}(\tilde{x}, \lambda^{(k)}) = z_{LD}(\tilde{x}) \le z_{MILP}^*. \tag{5}$$

The optimal solution $\tilde{x}^*$ of the Lagrangian relaxation $z_{\text{Lagrangian}}(\tilde{x}, \lambda)$ typically satisfies most original constraints and provides valuable structural information for the original problem, which can guide MILP solvers by defining high-quality search regions.

## 4 OUR METHOD

We propose **VRG**, a Lagrangian relaxation-guided score generation framework in visual space for MILP problem solving. VRG reshapes the solution vector so that each variable is represented as a pixel in an image. A U-Net-based convolutional network then operates on this "MILP-image" to capture complex inter-variable correlations, enabling more accurate score estimation over the solution distribution. To bias generation toward feasible and high-quality solutions, VGR integrates Lagrangian relaxation directly into a score-based generative process: constraint and objective information act as guidance terms during sampling. This integration preserves the multimodal nature of diffusion (diverse candidates) while providing guided trajectories toward feasible, near-optimal regions, effectively addressing key limitations of prior prediction-based approaches.

## 4.1 Visual-Vector Transformation

To capture structural relationships among variables, we transform a solution vector into an image. Given a MILP solution vector $x \in \mathbb{R}^n$, we reshape the vector into a 2D grid of dimensions $h \times w$, referred to as the **MILP-image**. Let $w = \lceil \frac{n}{h} \rceil$, and define the MILP-image as:

$$\mathcal{X} = [\mathcal{I}] \in \{[0,1]\}^{1 \times h \times w} \tag{6}$$

where $\mathcal{I}_{i,j} = x_{(i-1)\,w+j}$. Following this conversion, we define the transformation function from visual to vector as:

$$x = \mathbf{Vec}(\mathcal{X}) \in \mathbb{R}^n, \tag{7}$$

It is straightforward that this visual-to-vector mapping **Vec** is lossless and bijective under the assumption $h \times w = n$. Hence, $\mathbf{Vec}^{-1}$ exists, and we can transform the $x$ to $\mathcal{X}$ by $\mathbf{Vec}^{-1}$. Normalization is applied to ensure all values lie in $[0,1]$, using an invertible transform that preserves bijectivity. After this bijective conversion, the visual form, combined with convolutional U-Net architectures, naturally captures connections between pixel blocks (i.e., groups of variables) corresponding to solution components. This enhances the model's capacity to approximate probability distributions with rich inter-variable coupling and leads to more precise score estimation for subsequent modeling.

## 4.2 Lagrangian Relaxation-guided SDEs

Given an optimal solution $x^*$ and its corresponding MILP instance $g$, we aim to approximate the visual-instance distribution $p(\mathcal{X}|g)$. Following the typical MILP bipartite graph representation with constraint and variable nodes (Gasse et al., 2019), any MILP instance can be encoded into a graph embedding $\mathbf{g}$ using a 2-layer GNN. Under this encoding, we assume $p(x^*|g) \cong p(x^*|\mathbf{g})$.

With the bijection, we likewise identify:

$$p(\mathcal{X}^*|\mathbf{g}) \cong p(x^*|g) \tag{8}$$

However, the distribution $p(\mathcal{X}^*|\mathbf{g})$ remains purely date-driven and lacks **explicit optimality and feasibility guidance**. As illustrated in Fig. 2(a), sampling from this distribution may produce low-quality or infeasible solutions. Leveraging insights from Theorem **??**, which guarantees that Lagrangian relaxed solutions exhibit high feasibility and optimality for MILP problems, we propose to inject Lagrangian relaxation into the diffusion-based probabilistic model as guidance.

For any candidate solution $\tilde{x}$, we define two regularized terms for Lagrangian relaxation function: $\mathcal{O}(\tilde{x}) = |c^T(\tilde{x} - x^*)|$, which mea-

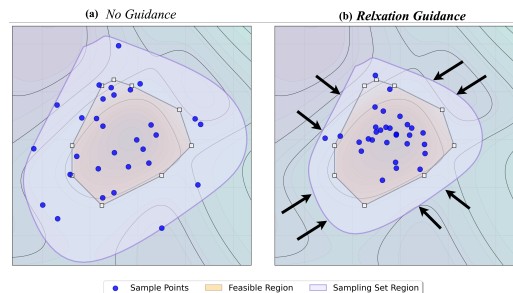

Figure 2: (a) Sampling without guidance. (b) Sampling with Lagrangian relaxation guidance, which biases sampling toward regions with **higher feasibility and optimality**.

sures the distance from the optimal solution (optimality), and $\mathcal{P}(\tilde{x}) = \|\max\{A\tilde{x} - b, 0\}\|$, which aggregates constraint violations (feasibility). Classical Lagrangian penalties can become unbounded on infeasible solutions, which is incompatible with probabilistic models that require finite probability mass and stable gradients. Thus, we reformulate the original objective into a regularized term with bounded penalties: $\gamma_o \mathcal{O}(\tilde{x}) + \gamma_c \mathcal{P}(\tilde{x})$ where $\gamma_o, \gamma_c \in \mathbb{R}^+$. These penalties guide sampling toward optimal and feasible regions while remaining compatible with continuous probabilistic models (no infinite penalties), such as diffusion or score-based SDEs. Let $\mathcal{X}^* = \mathbf{Vec}^{-1}(x^*)$.

We then aim to learn a probability distribution $q(x|\mathbf{g})$ that approximates $p(x^*|\mathbf{g})$ while minimizing the regularized Lagrangian objective. We prove that this optimization process equals to a KL divergence minimization against a guided distribution $\tilde{p}(x^*|\mathbf{g})$ in Theorem 2,which indicates that we can seamlessly integrate this guidance into diffusion modeling.

**Theorem 2** (Optimization Equivalence). *Let $\gamma_o, \gamma_c \in \mathbb{R}^+$ be bounded. Then, minimizing the regularized objective:*

$$\min_q [D_{KL}(q(x|\mathbf{g})\|p(x^*|\mathbf{g})) + \gamma_o \mathcal{O}(x) + \gamma_c \mathcal{P}(x)] \tag{9}$$

*is equivalent to minimizing the KL divergence with respect to the guided distribution $\tilde{p}(x|\mathbf{g})$:*

$$\min_q D_{KL}\left(q(x|\mathbf{g})\|\tilde{p}(x|\mathbf{g})\right) \tag{10}$$

*where $\tilde{p}(x|\mathbf{g}) \cong \frac{1}{Z}p(x^*|\mathbf{g})\exp\left(-\gamma_o\mathcal{O}(x) - \gamma_c\mathcal{P}(x)\right)$ and $Z$ is the normalization constant.*

Proof is provided in Appendix A. Based on this, and using the bijection $x = \mathbf{Vec}(\mathcal{X})$, we define the Lagrangian-guided target distribution over the visual space as(combined with $p(x^*|\mathbf{g}) \cong p(\mathcal{X}^*|\mathbf{g})$):

$$\tilde{p}(\mathcal{X}|\mathbf{g}) = p(\mathcal{X}^*|\mathbf{g})\exp\left(-\gamma_o\mathcal{O}(\mathbf{Vec}(\mathcal{X})) - \gamma_c\mathcal{P}(\mathbf{Vec}(\mathcal{X}))\right) \tag{11}$$

Theorem 3 (proof in Appendix B) theoretically guarantees that this Lagrangian-guided distribution effectively reshapes the probability landscape: for any $0 < \alpha < 1$, the top-$\alpha$ high-probability region under $\tilde{p}$ is a subset of that under $p$. Consequently, probability mass is shifted away from low-quality solutions and concentrated around high-quality, feasible regions, resulting in a tighter search space and more efficient sampling.

**Theorem 3** (**VRG** Tighter Solution Theorem). *Let $\Omega_1 = \{\mathcal{X} : p(\mathcal{X}|\mathbf{g}) \geq \alpha\max_{\mathcal{X}'}p(\mathcal{X}'|\mathbf{g})\}$ and $\Omega_2 = \{\mathcal{X} : \tilde{p}(\mathcal{X}|\mathbf{g}) \geq \alpha\max_{\mathcal{X}'}\tilde{p}(\mathcal{X}'|\mathbf{g})\}$ for some $0 < \alpha < 1$. Then $\Omega_2 \subseteq \Omega_1$, indicating that Lagrangian guidance concentrates probability mass on higher-quality solutions.*

To get the Lagrangian-guided target distribution $\tilde{p}(\mathbf{Vec}(\mathcal{X}))$,we construct a reverse-time Stochastic Differential Equation (SDE) whose stationary distribution converges to $\tilde{p}$, we derive the corresponding reverse process from the gaussian distribution $\tilde{p}_T \sim N(0, I)$ and integrate the Lagrangian regularization guidance to the drift, as in Theorem 4.

**Theorem 4** (*Relaxation-guided SDE with Score Function*). *Given the Lagrangian-guided $\tilde{p}(x|\mathbf{g})$, the corresponding reverse-time SDE that samples from $\tilde{p}$ is given by:*

$$d\mathcal{X}_t = \left[-\frac{1}{2}\beta(t)\mathcal{X}_t - (\beta(t)\partial_{\mathcal{X}}p(\mathcal{X}^*|\mathbf{g}) - \beta(t)\gamma_o\partial_x\mathcal{O}(x_t) - \beta(t)\gamma_c\partial_x\mathcal{P}(x_t))\right]dt + \sqrt{\beta(t)}d\bar{W}_t \tag{12}$$

*where $x = \mathbf{Vec}(.)$, $x_t = \mathbf{Vec}(\mathcal{X}_t)$, $\partial_{\mathcal{X}}p(\mathcal{X}^*|\mathbf{g})$ is the approximated score of the data distribution, $d\bar{W}_t$ is a standard Wiener process, and $\beta(t)$ is the noise schedule. This SDE integrates optimization-aware gradient signals into the diffusion process, guiding trajectories toward feasible and optimal regions during sampling.*

*The overall score function implied by Eq. 12, and used in both training and sampling, is:*

$$\boxed{s_\theta^*(\mathcal{X}_t, t, \mathbf{g}) = \partial_{\mathcal{X}}p(\mathcal{X}^*|\mathbf{g}) - \gamma_o\partial_x\mathcal{O}(x_t) - \gamma_c\partial_x\mathcal{P}(x_t)} \tag{13}$$

Proof is provided in Appendix C.

To compute $\mathcal{P}(x)$, we define $\mathcal{P}(x) = \sum_i^m \mathcal{P}_i(x)$, $\mathcal{P}_i(x) = \text{ReLU}((Ax)_i - b_i)) \in \mathbb{R}$ so that $\mathcal{P}(x)$ measures the aggregated violation of the linear constraint $(Ax)_i \leq b_i$. If $x \in D$ (the feasible region), then $\mathcal{P}(x) = 0$; otherwise $\mathcal{P}(x) > 0$. The gradient $\partial_x\mathcal{P}(x)$ provides a direction to reduce constraint violations during both training and sampling.

## 4.3 Training and Sampling

**Conditional Score Network.** To train the score function in Eq.13, we introduce a conditional score network that incorporates the MILP structure as context. Specifically, we first encode an MILP instance with a GNN encoder $\tau_\phi$ to produce an instance embedding $\mathbf{g} = \tau_\phi(A, b, c) \in \mathbb{R}^{d_{\text{cond}}}$.The score model is a conditional U-Net style network $s_\theta(x_t, t, \mathbf{g})$ that takes the noisy image input $\mathcal{X}_t$ at time $t$ and injects the MILP structure $\mathbf{g}$ via cross-attention(Rombach et al., 2022; Saharia et al., 2022). The attention mechanism is defined as:

$$Q = W_Q\mathcal{X}_T, \quad K = W_K\mathbf{g}, \quad V = W_V\mathbf{g}, \quad \mathcal{X}_t = \mathcal{X}_t + \text{Attention}(Q, K, V)$$
$$\text{where} \quad \text{Attention}(Q, K, V) = \text{softmax}\left(\frac{QK^{\top}}{\sqrt{d_k}}\right)V. \tag{14}$$

with learnable projection matrices $W_Q \in \mathbb{R}^{d_{\text{img}} \times d_k}$, $W_K \in \mathbb{R}^{d_{\text{cond}} \times d_k}$, and $W_V \in \mathbb{R}^{d_{\text{cond}} \times d_v}$. $d_k = d_v = \frac{d_{\text{cond}}}{H}$ for multi-head attention with $H$ attention heads.

**Training with Lagrangian-guided score.** To learning the Lagrangian-relaxation guided score defined in Theorem 4, we minimize the following training objective:

$$L_{\text{Relaxed}} = \mathbb{E}\|s_\theta(\mathcal{X}_t, t, \mathbf{g}) - s_{\text{target}}\| \tag{15}$$

where the target score is:

$$\begin{aligned} s_{\text{target}} &= s_\theta^*(\mathcal{X}_t, t, \mathbf{g}) = \partial_\mathcal{X} p(\mathcal{X}^*|\mathbf{g}) - \gamma_o \partial_x \mathcal{O}(x_t) - \gamma_c \partial_x \mathcal{P}(x_t) \\ &= \partial_\mathcal{X} p(\mathcal{X}^*|\mathbf{g}) - \gamma_o \partial_x \mathcal{O}(\mathbf{Vec}(\mathcal{X}_t)) - \gamma_c \partial_x \mathcal{P}(\mathbf{Vec}(\mathcal{X}_t)) \end{aligned} \tag{16}$$

Following Song et al. (2021), we corrupt the clean solution $\mathcal{X}_0$ into a noisy version $\mathcal{X}_t$ using the standard forward process:

$$\mathcal{X}_t = \sqrt{\bar{\alpha}_t}\,\mathcal{X}_0 + \sqrt{1-\bar{\alpha}_t}\,\epsilon, \quad \epsilon \sim \mathcal{N}(0, I). \tag{17}$$

Using the chain rule, we derive the gradient of the Lagrangian terms as:

$$\partial_x \mathcal{O}(\mathbf{Vec}(\mathcal{X}_t)) = \frac{\partial \mathcal{O}(\mathbf{Vec}(\mathcal{X}_t))}{\partial \mathbf{Vec}(\mathcal{X})} = \frac{\partial \mathcal{O}(\mathbf{Vec}(\mathcal{X}_t))}{\partial \mathbf{Vec}(\mathcal{X}_t)} \cdot \frac{\partial \mathbf{Vec}(\mathcal{X}_t)}{\partial \mathbf{Vec}(\mathcal{X})} \tag{18}$$

As the objective deviation is defined as $\mathcal{O}(\tilde{x}) = |c^T(\tilde{x} - x^*)|$, applying Eq.17 yields:

$$\frac{\partial \mathcal{O}(\mathbf{Vec}(\mathcal{X}_t))}{\partial \mathbf{Vec}(\mathcal{X}_t)} = c^T \cdot \text{sign}(\mathbf{Vec}(\mathcal{X}_t) - \mathbf{Vec}(\mathcal{X}^*)) \quad \text{and} \quad \frac{\partial \mathbf{Vec}(\mathcal{X}_t)}{\partial \mathbf{Vec}(\mathcal{X})} = \sqrt{\bar{\alpha}_t} \cdot I \tag{19}$$

Combining Eq 18 and Eq 19, $\partial_x \mathcal{O}(\mathbf{Vec}(\mathcal{X}_t)) = c\sqrt{\bar{\alpha}_t} \cdot \text{sign}(\mathbf{Vec}(\mathcal{X}_t) - \mathbf{Vec}(\mathcal{X}^*))$. Similarly, for the constraint penalty term, we can get $(\partial_{\mathcal{X}_t}(\lambda) = 0)$: $\partial_x \mathcal{P}(\mathbf{Vec}(\mathcal{X}_t)) = \sqrt{\bar{\alpha}_t}\lambda^T A \mathbf{Vec}(\mathcal{X}_t)$. Substituting these into the score matching loss leads to the simplified objective:

$$L_{\text{simplified}} = \mathbb{E}\|s_\theta(\mathcal{X}_t, t, \mathbf{g}) - (\epsilon_t - \gamma_o c\sqrt{\bar{\alpha}_t} \cdot \mathbf{Vec}(|\mathcal{X}_t - \mathcal{X}^*|) - \gamma_c \sqrt{\bar{\alpha}_t}\lambda^T A \mathbf{Vec}(\mathcal{X}_t))\| \tag{20}$$

where $\epsilon_t \sim \mathcal{N}(0, I)$ following the standard score-based generation model (Song et al., 2021). This objective retains the standard diffusion target $\epsilon_t$ and adds optimization-aware corrections: an objective-gap term along $c$ and a feasibility term along $\lambda^T A$, both scaled by $\sqrt{\bar{\alpha}_t}$.

**Guided Sampling.** After training, we sample using the learned score model $s_\theta(.)$. Specifically, we initialize from a Gaussian prior $\tilde{p}_T \sim N(0, I)$ at visual space (Ho et al., 2020) and use the trained score function to reach the guided distribution $\tilde{p}(\mathcal{X}|\mathbf{g})$, which captures both feasibility and near-optimality. Then sample $\tilde{\mathcal{X}}_0$ from the $\tilde{p}(\mathcal{X}|\mathbf{g})$ and then go back to the solution space by $\mathbf{Vec}$. The whole process can be explained in the Algorithm 1

**Bounded Relaxation Convergence.** Given the mild regularity assumptions (Assumptions D.1 and D.2), we prove that the Lagrangian relaxation-guided sampling admits a finite approximation error bound. Specifically, Theorem 5 demonstrates that the VRG probabilistic generative model converges to the optimal solution with a quantifiable optimality gap, provided that the solution domain is bounded and the score network approximation error is controlled. The complete proof is provided in Appendix D.

---

**Algorithm 1** Relaxation Guided Sampling

**Notation:** $s_\theta$ is the Lagrangian relaxation guided score function trained by $L_{\text{simplified}}$.

1: $\mathcal{X}_T \sim \mathcal{N}(\mathbf{0}, \mathbf{I})$ # Initialize using Gaussian noise in visual space
2: **for** $t = T, T-1, \ldots, 1$ **do**
3:     $s \leftarrow s_\theta(\mathcal{X}_t, t, \mathbf{g})$
4:     $\boldsymbol{\mu}_t \leftarrow \frac{1}{\sqrt{\alpha_t}}\left(\tilde{\mathcal{X}}_t - \frac{\beta_t}{\sqrt{1-\bar{\alpha}_t}}s\right)$
5:     **if** $t > 1$ **then**
6:        $\mathbf{z} \sim \mathcal{N}(\mathbf{0}, \mathbf{I})$
7:        $\tilde{\mathcal{X}}_{t-1} \leftarrow \boldsymbol{\mu}_t + \sqrt{\beta_t}\mathbf{z}$
8:     **else**
9:        $\tilde{\mathcal{X}}_0 \leftarrow \boldsymbol{\mu}_t$
10:     **end if**
11: **end for**
12: $\tilde{x} = \mathbf{Vec}(\tilde{\mathcal{X}}_0)$ # Go back to solution space
13: **return** $\tilde{x}$

---

To fully exploit the multi-modal nature of the guided distribution, we adopt a parallel sampling strategy . For each MILP instance: 1) Sample $k$ visual solution candidates $\{\tilde{\mathcal{X}}^j, j = 1, 2..k\}$ using different random seeds in image space. 2) Map each candidate to vector form $\tilde{x}_j = \mathbf{Vec}(\tilde{\mathcal{X}}_j)$. That is to say, steps 1) and 2) are concurrent multi-threaded parallel executions of Algorithm 1. 3) Select the best solution $\tilde{x}^*$ using the metrics $\mathcal{O}(\cdot)$ and $\mathcal{P}(\cdot)$. This strategy enhances robustness by exploring diverse high-probability regions in the solution space before downstream search/polishing.

## 4.4 SEARCH PROCEDURE

Given a generated solution $\tilde{x}^*$, we construct a trust region constraint to reduce problem complexity and refine solution quality. Unlike PaS (Han et al., 2023), which relies on fixed 0-1 constraint tuning, our approach leverages information from the relaxed objective and constraint values to shape the search space. Specifically, we project the integer-valued subvector of $\tilde{x}^*$ to obtain $\tilde{x}_I$, and then define a trust region around it. The resulting search subproblem is formulated as:

$$M_{\text{search}}(\tilde{x}_I) : \min_{x \in D \cap B'(\tilde{x}_I, \Delta)} c^T x, \tag{21}$$

where $B'(\tilde{x}_I, \Delta) \equiv \{x \in \mathbb{R}^n : \|\tilde{x}_I - x_I\|_1 \leq \Delta\}$ denotes the trust region, $\Delta$ controls the allowable deviation. We solve this subproblem with industry-standard MILP solvers, such as SCIP and Gurobi, following the established setup in prior work (Han et al., 2023), to get the solved solution $\hat{x}^*$.

# 5 EXPERIMENTS

## 5.1 SETTINGS

**Dataset Benchmark.** We evaluate on four widely used NP-hard public MILP benchmarks: 1) *Set Covering (SC)* (Chvátal, 1979); 2) *Combinatorial Auction (CA)* (Sandholm, 1999); 3)*Capacitated Facility Location (FC)* (Balinski, 1965); and 4) *Maximum Independent Set (IS)* (Karp, 1972b). Each benchmark is provided at three scales, Small, Medium, and Large, based on the number of variables and constraints. The details are summarized in Table 9 in Appendix.

**Baselines.** We compare against two exact expert MILP solvers and two end-to-end learning-based methods: 1) *SCIP* (Bestuzheva et al., 2023): a leading open-source solver. 2) *Gurobi* (Gurobi Optimization, LLC, 2023): a state-of-the-art commercial MILP solver. 3) *Predict-and-Search (PaS)* Han et al. (2023): a learning-based method that predicts good solutions and performs local search. 4) *Contrastive Predict-and-Search (CoPaS)* (Gao et al., 2024): an improved PaS with contrastive learning to enhance solution quality. We also introduce two advanced ML-based MILP solving baselines: 5) Apollo-MILP (Geng et al., 2025) and 6) DiffILO (Liu et al., 2025)

**Evaluation Metrics.** We report four metrics: 1) Obj: The best optimal objective value by $c\hat{x}^*$. 2) Gap (%): The optimality gap of the solved optimal objective by $c^T \hat{x}^*$ relative to the best known solution (BKS), Gap $= \frac{c^T \hat{x}^* - \text{BKS}}{\text{BKS}}$ for minimization problems (e.g, SC and FC), and reversed accordingly for maximization problems (e.g., IS and CA). 3) Searching Time (s): Time to find the optimal solution from the constructed subproblem $M_{\text{search}}(\tilde{x}_I)$, starting from the predicted solution $\hat{x}^*$. 4) Nodes: The number of branch-and-bound nodes explored by the solver to reach the final solution, reflecting computational effort.

**Hyperparameters.** For learning-based solvers, we use a consistent search radius $\Delta$ (details in Appendix E.2). For our **VRG**, we specify the **MILP-image** resolution, sampling steps, and the # of samples per benchmark scale. Full implementation details are provided in Appendix E.3 and E.5.

## 5.2 MAIN RESULTS

**Benchmark Solving Results** We evaluate all methods on the four benchmarks at three scales. Objective values (Obj) and optimality gaps (Gap) are reported in Table 1, 2, and 3. Computational efficiency in terms of Searching Time (s) and Nodes is reported in Table 5, using SCIP and Gurobi as reference solvers.

In terms of quality, our method **VRG** consistently outperforms learning-based baselines across all benchmarks and scales. When paired with SCIP or Gurobi, **VRG** achieves the lowest optimality gaps and often matches the exact solvers (e.g., SCIP, Gurobi) while maintaining feasibility. Regarding search efficiency, VRG significantly reduces both search time and the number of explored nodes (Table 5), demonstrating its ability to guide solvers toward high-quality regions. For robustness, VRG avoids infeasible subproblem construction,some failure in PaS-based methods (see $\infty$ entries). This improvement stems from the multi-modal, feasibility-aware sampling mechanism of our method, which generates diverse, high-quality candidate solutions that remain within the feasible region. These candidates, in turn, enable more effective local search under trust-region constraints.

Table 1: Results on small-scale MILP benchmarks. Results are averaged over 100 test instances. "$\infty$" indicates subproblem $M_{\text{search}}(\tilde{x})$ is infeasible. All solvers have a 1000-second time limit.

| Method | Style | SC(BKS = 21.17) | | MIS(BKS = 3529.01) | | CA(BKS = 13129.78) | | FC(BKS = 6879.07) | |
|---|---|---|---|---|---|---|---|---|---|
| | | Obj($\downarrow$). | Gap($\downarrow$)(%) | Obj($\uparrow$). | Gap($\downarrow$)(%) | Obj($\uparrow$). | Gap($\downarrow$)(%) | Obj($\downarrow$). | Gap ($\downarrow$)(%)(s) |
| SCIP | Exact | 21.17 | 0.0000 | 3542.69 | -0.3876 | 13129.78 | 0.0000 | 6879.07 | **0.0000** |
| Gurobi | Exact | 21.17 | 0.0000 | 3529.01 | 0.0000 | 13129.78 | 0.0000 | 6791.36 | -1.2750 |
| PaS + SCIP | Pre. | 78.40 | 261.2903 | 3538.03 | -0.2556 | 12741.15 | 2.9599 | 3359.89 | 51.1577 |
| PaS + Gurobi | Pre. | 78.40 | 261.2903 | 3523.13 | 0.1666 | 12730.90 | 3.0379 | 3359.89 | 51.1577 |
| CoPaS + SCIP | Pre. | 27.99 | 32.2153 | 2712.00 | 23.1512 | 12259.15 | 6.6309 | $\infty$ | $\infty$ |
| CoPaS + Gurobi | Pre. | 27.99 | 32.2153 | 2697.01 | 2357.54 | 12667.78 | 1.7365 | $\infty$ | $\infty$ |
| **VRG+ SCIP (Ours)** | Gen. | **21.17** | **0.0000** | 3542.69 | -0.3876 | 13129.78 | 0.0000 | 6879.07 | 0.0000 |
| **VRG + Gurobi (Ours)** | Gen. | 21.17 | 0.0000 | 3529.01 | 0.0000 | **13129.78** | **0.0000** | **6791.36** | **-1.2750** |

Table 2: Results on medium-scale MILP benchmarks. All solvers have a 1000-second time limit and the results are averaged over 20 instances.

| Method | Style | SC(BKS = 31.30) | | MIS(BKS = 7185.65) | | CA(BKS = 64552.19) | | FC(BKS = 11344.00) | |
|---|---|---|---|---|---|---|---|---|---|
| | | Obj($\downarrow$). | Gap($\downarrow$)(%) | Obj($\uparrow$). | Gap($\downarrow$)(%) | Obj($\uparrow$). | Gap($\downarrow$)(%) | Obj($\downarrow$). | Gap($\downarrow$)(%) |
| SCIP | Exact | 31.30 | 0.0000 | 7185.65 | 0.0000 | 71247.97 | –10.3726 | 9517.42 | -16.1017 |
| Gurobi | Exact | 31.30 | 0.0000 | 7185.65 | 0.0000 | 71247.97 | –10.3726 | 9450.79 | -16.6890 |
| PaS + SCIP | Pre. | 475.70 | 1400.6389 | 7158.94 | 0.3717 | 70757.07 | -9.6121 | 9716.22 | -14.3492 |
| PaS + Gurobi | Pre. | 475.70 | 1400.6389 | 7132.50 | 0.7396 | 70731.56 | -9.5568 | 9601.89 | -15.3571 |
| CoPaS + SCIP | Pre. | 32.40 | 3.5143 | 6365.50 | 11.4137 | 65611.50 | -1.6410 | 9517.42 | -16.1017 |
| CoPaS + Gurobi | Pre. | 32.40 | 3.5143 | 6365.50 | 11.4137 | 65611.50 | -1.6410 | 9450.79 | -16.6890 |
| **VRG+ SCIP (Ours)** | Gen. | 31.30 | 0.0000 | **7203.89** | **-0.2538** | 71247.97 | -10.3726 | 9517.42 | -16.1017 |
| **VRG+ Gurobi (Ours)** | Gen. | **31.30** | 0.0000 | 7185.65 | 0.0000 | **71247.97** | **-10.3726** | 9450.79 | -16.6890 |

## 5.3 GENERATION ANALYSIS

**Visualization of Generation Process.** We visualize the sampling process across problem instances of varying scales. Fig. 3 (Appendix F) illustrates generation trajectories for medium- and large-scale problems. Throughout the generation process, we observe a consistent, monotonic decrease in both Lagrangian regularization terms $\mathcal{O}(\mathbf{Vec}(\tilde{\mathcal{X}}_t))$ and $\mathcal{P}(\mathbf{Vec}(\tilde{\mathcal{X}}_t))$ for images sampled at step $t$. This empirical observation validates our model's progressive refinement capability, demonstrating its structured guidance iteratively improves solution quality and ultimately converges to high-quality feasible solutions.

Table 3: Results on large-scale MILP benchmarks. All solvers have a 1000-second time limit and the results are averaged over 20 instances.

| Method | Style | SC(BKS = 26.20) | | MIS(BKS = 7132.50) | | CA(BKS = 120914.75) | | FC(BKS = 11552.13) | |
|---|---|---|---|---|---|---|---|---|---|
| | | Obj($\downarrow$). | Gap($\downarrow$)(%) | Obj($\uparrow$). | Gap($\downarrow$)(%) | Obj($\uparrow$). | Gap($\downarrow$)(%) | Obj($\downarrow$). | Gap($\downarrow$)(%) |
| SCIP | Exact | 26.20 | 0.0000 | 7132.50 | 0.0000 | 120914.75 | 0.0000 | 11552.13 | 0.0000 |
| Gurobi | Exact | 25.80 | -1.2567 | 7132.50 | 0.0000 | 120914.75 | 0.0000 | 11547.33 | -0.0415 |
| PaS + SCIP | Pre. | 26.00 | -0.7633 | 7132.22 | 0.0039 | 120914.75 | 0.0000 | $\infty$ | $\infty$ |
| PaS + Gurobi | Pre. | 26.00 | -0.7633 | 7109.40 | 0.3238 | 120925.61 | -0.0089 | $\infty$ | $\infty$ |
| CoPaS + SCIP | Pre. | 30.50 | 22.6640 | 6364.50 | 10.7676 | 120914.75 | 0.0000 | 11552.13 | 0.0000 |
| CoPaS + Gurobi | Pre. | 30.50 | 22.6640 | 6364.50 | 10.7676 | 120914.75 | 0.0000 | 11552.13 | 0.0000 |
| **VRG + SCIP (Ours)** | Gen. | 25.80 | -1.2567 | **7144.22** | **-0.1643** | 120914.75 | 0.0000 | 11552.14 | 0.0000 |
| **VRG + Gurobi (Ours)** | Gen. | **25.80** | **-1.2567** | 7132.50 | 0.0000 | **120945.43** | **-0.0253** | **11546.33** | **-0.0415** |

**Multi-modal Sampling.** We further examine the multi-modal nature of our generation process. Fig. 4 and 6 in Appendix F.2 confirm that parallel sampling operations in our framework lead to diverse visual transformations with distinct modal characteristics. This diversity increases the chance of generating high-quality subproblems when transforming from the sampled space to the solution space, enhancing both stability and robustness in the final search and consistently guiding the solver toward feasible, high-quality regions.

**Generation Time Analysis.** We compare generation time (solution construction before solver search) across all benchmarks with different scales. Table 4 shows that our method introduces min-

imal computational overhead relative to solver search time. Specifically, in 9 out of 12 benchmark settings, sampling takes under 0.4 seconds, less than 0.04% of the total allocated runtime (1000 seconds). This negligible cost demonstrates the practical efficiency of our method, which can generate numerous high-quality solutions that are both feasible and near-optimal, even for large-scale MILPs.

Table 4: Generation time for **VRG** on medium and large benchmarks. Time ratios $\eta_S(\%)$ and $\eta_G(\%)$ are relative to the search time of SCIP and Gurobi, respectively. Reported time includes only visual generation process.

| | SC | | | MIS | | | FC | | | CA | | |
|---|---|---|---|---|---|---|---|---|---|---|---|---|
| | Time(s) | $\eta_S$ (%) | $\eta_G$ (%) | Time(s) | $\eta_S$ (%) | $\eta_G$ (%) | Time(s) | $\eta_S$ (%) | $\eta_G$ (%) | Time(s) | $\eta_S$ (%) | $\eta_G$ (%) |
| Small | 0.4116 | 2.9004 | 15.5985 | 0.2375 | 3.3928 | 77.0353 | 0.4953 | 19.5778 | 107.7676 | 0.2666 | 6.9320 | 101.7169 |
| Medium | 0.2296 | 0.1768 | 1.6874 | 0.2374 | 0.1687 | 3.3586 | 0.2083 | 87.6683 | 56.1304 | 0.2401 | 0.3205 | 4.2291 |
| Large | 0.2344 | 0.0234 | 0.0343 | 0.2340 | 0.2362 | 9.3926 | 0.2352 | 158.5974 | 160.5460 | 0.2556 | 0.0256 | 0.0285 |

Table 5: Searching time (s) and Nodes for **VRG**-based solvers vs. exact solvers across small, medium, and large benchmarks.

| | SC(Small) | | MIS(Small) | | FC(Small) | | CA(Small) | |
|---|---|---|---|---|---|---|---|---|
| | Time($\downarrow$) | Nodes($\downarrow$) | Time($\downarrow$) | Nodes($\downarrow$) | Time($\downarrow$) | Nodes($\downarrow$) | Time($\downarrow$) | Nodes($\downarrow$) |
| SCIP | 14.1911 | 104.4 | 7.0001 | 33.2 | 2.5105 | 1.0 | 3.8459 | 7.7 |
| **VRG + SCIP** | **7.1347** | **76.96** | **4.6093** | **33.2** | 2.5702 | 1.0 | **3.4765** | **7.7** |
| Gurobi | 2.6387 | **312.9** | 0.3083 | 4.4 | 0.4596 | 1.0 | 0.2621 | 114.8 |
| **VRG + Gurobi** | **1.2686** | 318.5 | 0.3021 | 5.7 | **0.2868** | 1.0 | **0.2489** | **113.2** |
| | SC(Medium) | | MIS(Medium) | | FC(Medium) | | CA(Medium) | |
| | Time($\downarrow$) | Nodes($\downarrow$) | Time($\downarrow$) | Nodes($\downarrow$) | Time($\downarrow$) | Nodes($\downarrow$) | Time($\downarrow$) | Nodes($\downarrow$) |
| SCIP | 129.7983 | 7585.5 | **140.6765** | 115.7 | **0.2376** | 1.0 | 74.9119 | 2845.2 |
| **VRG + SCIP** | **112.8781** | **7446.4** | 191.1203 | **102.3** | 0.3032 | **1.0** | **73.5590** | **2821.0** |
| Gurobi | 13.6067 | 10894.5 | 7.0684 | 53.9 | 0.3711 | 6.6 | 5.6915 | 6137.9 |
| **VRG+ Gurobi** | **13.2742** | **10453.8** | **7.0176** | **53.0** | **0.3588** | **5.2** | **5.0330** | **5701.2** |
| | SC(Large) | | MIS(Large) | | FC(Large) | | CA(Large) | |
| | Time($\downarrow$) | Nodes($\downarrow$) | Time($\downarrow$) | Nodes($\downarrow$) | Time($\downarrow$) | Nodes($\downarrow$) | Time($\downarrow$) | Nodes($\downarrow$) |
| SCIP | 1000.0079 | **9319.60** | 99.0448 | 165.0 | 0.1483 | 1.0 | 1000.01 | 22215.8 |
| **VRG + SCIP** | **1000.0065** | 10245.2 | **54.3102** | **162.6** | **0.0725** | 1.0 | **1000.00** | **7792.7** |
| Gurobi | 683.2202 | 598502.0 | 4.0207 | **28.1** | 0.1465 | 1.0 | **897.4849** | **756520.9** |
| **VRG + Gurobi** | **623.9327** | **456737.8** | **2.3841** | 37.0 | **0.1440** | 1.0 | 899.0478 | 777839.2 |

## 5.4 ABLATION STUDY

**Ablation on the relaxation guidance.** To assess the effectiveness of our relaxation guidance mechanism, we conduct ablation studies on two representative MILP problems: large-scale Maximum Independent Set (MIS) and small-scale Capacitated Assignment (CA). The goal is to evaluate the impact of relaxation guidance on solver performance regarding search time. We disable the relaxation guidance component during training and sampling, while keeping all other settings identical. The results in Table 6 reveal a substantial increase in searching time when guidance is removed, confirming the critical role of Lagrangian guidance in steering the solver toward high-quality regions and improving search efficiency.

**Impact of Image Resolution** $(h, w)$ **and Guidance Weights** $\gamma_o, \gamma_c$**.** We evaluate the sensitivity of VRG to image resolution. The results in Table 7 and Figure 7 demonstrate that image resolution has minimal impact on solving performance, even with extreme aspect ratios such as $(h, w) = (2, 250)$ and $(5, 100)$. We also visualize the generation process from 20% to 80% completion, observing that the generated visual representation $\mathcal{X}_T$, under the same guidance weights $\gamma_o$ and $\gamma_c$, consistently achieves low constraint violations and low objective values for minimization tasks in Figure 9 from Appendix (e.g., SC). Regarding the guidance weights $\gamma_o$ and $\gamma_c$, our findings suggest that, for SC problems,

Table 6: Ablation study on relaxation guidance.

| Method | IS(SCIP) Time($\downarrow$) | IS(Gurobi) Time($\downarrow$) |
|---|---|---|
| **VRG w/o Guidance** | 178.9089 | 3.3546 |
| **VRG (Ours)** | **54.3102** | **2.3841** |
| **Method** | CA (SCIP) Time($\downarrow$) | CA (Gurobi) Time($\downarrow$) |
| **VRG w/o Guidance** | 6.4299 | 0.4223 |
| **VRG (Ours)** | **4.7252** | **0.1920** |

bounding these parameters within a moderate range (e.g., $\gamma_o, \gamma_c \in [1, 4]$) yields stable performance. Furthermore, when using different diffusion parameters $\alpha_0$ and $\beta_0$, larger values of $\gamma_o$ and $\gamma_c$ can be applied without degrading performance. In practice, extensive tuning of $\gamma_o$ and $\gamma_c$ is generally unnecessary, as the solving results remain robust across a wide range of settings.

**Ablation on Step Iterations.**
We assess the impact of generation steps on VRG's performance. As shown in the table 7, the performance variation across different generation step counts is minimal, indicating that our visual relaxation generative model is insensitive to the choice of generation steps during training.

**Ablation Study on Visual Encoder and Representation.** We evaluate the impact of different architectures on VRG's performance. We define the error metric $\sigma = |s_\theta - s_{\text{target}}|$ to quantify the model's ability to approximate the target guidance score across different representations. As shown in Table 8, when replacing the U-Net with an MLP for vector-based generation (while retaining our proposed guidance), the model fails to accurately estimate the target guidance score, yielding $\sigma \approx 1.2 \times 10^7$. This indicates that the MLP-based vector generation cannot capture the holistic structure of the guidance signal. Without the multi-scale receptive fields enabled by the convolutional architecture, the model loses the capacity to model interdependencies among variables, resulting in ineffective approximation of the guided score, which indicates the benefits from the visual representation.

|  | $(h, w)$ | $(\gamma_o, \gamma_c)$ | Step | SC(500,800) | |
|---|---|---|---|---|---|
|  |  |  |  | Nodes($\downarrow$) | Obj($\downarrow$) |
| Gurobi | – | – | – | $5841.46 \pm 5191.29$ | $28.9800 \pm 3.0270$ |
| SCIP | – | – | – | $3336.16 \pm 3846.81$ | $28.8936 \pm 3.0305$ |
| PaS + SCIP | – | – | – | $4873.59 \pm 5177.36$ | $28.9800 \pm 3.0117$ |
| DiffILO | – | – | – | $6252.90 \pm 7095.00$ | $28.9800 \pm 3.0270$ |
| Apollo MILP | – | – | – | $7707.42 \pm 7807.04$ | $28.9800 \pm 3.0270$ |
| VRG+ SCIP | (5,100) | (2,2) | 20 | $3020.22 \pm 3255.73$ | $28.8478 \pm 3.0475$ |
| VRG+ SCIP | (2,250) | (2,2) | 20 | $2994.58 \pm 3154.28$ | $\mathbf{28.7778} \pm 3.0442$ |
| VRG+ SCIP | (20,25) | (2,2) | 20 | $3309.38 \pm 3770.38$ | $28.9574 \pm 3.1065$ |
| VRG+ Gurobi | (5,100) | (2,2) | 20 | $5824.98 \pm 5635.45$ | $28.9800 \pm 3.0270$ |
| VRG+ Gurobi | (2,250) | (2,2) | 20 | $5824.98 \pm 5635.45$ | $28.9800 \pm 3.0270$ |
| VRG+ Gurobi | (20,25) | (2,2) | 20 | $5824.98 \pm 5635.45$ | $28.9800 \pm 3.0270$ |
| VRG+ SCIP | (20,25) | (2,1) | 20 | $2919.70 \pm 3319.68$ | $28.9574 \pm 3.1064$ |
| VRG+ SCIP | (20,25) | (1,1) | 20 | $\mathbf{2802.10} \pm 2843.02$ | $28.7778 \pm 3.0443$ |
| VRG+ SCIP | (20,25) | (1,2) | 20 | $2850.00 \pm 2789.88$ | $28.8913 \pm 3.1072$ |
| VRG+ Gurobi | (20,25) | (2,1) | 20 | $5824.98 \pm 5635.44$ | $28.9800 \pm 3.0270$ |
| VRG+ Gurobi | (20,25) | (1,1) | 20 | $5824.98 \pm 5635.44$ | $28.9800 \pm 3.0270$ |
| VRG+ Gurobi | (20,25) | (1,2) | 20 | $5824.98 \pm 5635.44$ | $28.9800 \pm 3.0270$ |
| VRG+ Gurobi | (20,25) | (3,4) | 20 | $5824.98 \pm 5635.44$ | $28.9800 \pm 3.0270$ |
| VRG+ Gurobi | (20,25) | (3,4) | 10 | $5824.98 \pm 5635.44$ | $28.9800 \pm 3.0270$ |
| VRG+ Gurobi | (20,25) | (3,4) | 5 | $5824.98 \pm 5635.44$ | $28.9800 \pm 3.0270$ |
| VRG+ SCIP | (20,25) | (3,4) | 20 | $\mathbf{2978.88} \pm 3074.71$ | $28.8913 \pm 3.1071$ |
| VRG+ SCIP | (20,25) | (3,4) | 10 | $3015.66 \pm 3161.26$ | $28.8913 \pm 3.1071$ |
| VRG+ SCIP | (20,25) | (3,4) | 5 | $3004.66 \pm 3130.51$ | $28.8913 \pm 3.1071$ |

Table 7: Comparison with other ML-based effective solvers using VRG models with different $\gamma_o, \gamma_c$ and $(h, w)$. Results are averaged over 50 test instances, each solved within 100s time limit. Step denotes the number of training steps.

## 6 CONCLUSION

In this work, we introduced **VRG**, a Lagrangian relaxation-guided score generation framework in visual space for Mixed Integer Linear Programming, to address key limitations in existing Predict-and-Search approaches. By transforming solution vectors into continuous image representations and leveraging a U-Net-based score network conditioned on instance structure, our method naturally captures variable interdependencies and generates diverse, high-quality candidate solutions. The Lagrangian relaxation guidance steers sampling toward feasible, near-optimal regions, yielding compact and effective trust-region subproblems for downstream

Table 8: Ablation study on visual representation and backbone layers (l) with $(\gamma_o, \gamma_c) = (1, 1)$, $(h, w) = (20, 25)$. Results are averaged over 40 instances.

| Method | Backbone | SC(500,800) |
|---|---|---|
|  |  | error($\downarrow$) |
| **VRG** w/o Visual | MLP | $> 1.2588 \cdot 10^7$ |
| **VRG** (Ours) | Unet. l=2 | 1.9107 |
| **VRG** (Ours) | Unet. l=3 | 1.9096 |
| **VRG** (Ours) | Unet. l=4 | 1.9106 |

solvers. Across public benchmarks, VRG consistently outperforms PaS-based baselines in solution quality. VRG also achieves competitive performance relative to state-of-the-art solvers such as SCIP and Gurobi with markedly lower computational effort (search time and explored nodes). These results suggest a promising path for integrating computer vision techniques with combinatorial optimization, enabling scalable and efficient MILP solving that preserves solution quality while enhancing computational efficiency.

REPRODUCIBILITY STATEMENT

Our work is fully reproducible. Complete source code is available through the anonymous repository: `https://anonymous.4open.science/r/VRG-E09E/`. All experimental configurations, hyperparameters, and implementation details are thoroughly documented in Appendix E. Additionally, we provide comprehensive instructions for environment setup and dataset preparation to facilitate replication of our results in above Appendix.

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

## A    PROOF OF THE EQUIVALENCE.

*Proof.* We prove the equivalence by showing that both optimization problems have the same objective function up to an additive constant.

Define the target distribution:

$$\tilde{p}(x|\mathbf{g}) = \frac{1}{Z} p(x^*|\mathbf{g}) \exp(-\gamma_o \mathcal{O}(x) - \gamma_c \mathcal{P}(x)) \tag{22}$$

where $Z = \int p(x^*|\mathbf{g}) \exp(-\gamma_o \mathcal{O}(x) - \gamma_c \mathcal{P}(x))dx$ is the normalization constant.

For the original optimization problem:

$$f_q = D_{\mathrm{KL}}(q(x)\|p(x^*|\mathbf{g})) + \gamma_o \mathbb{E}_{x \sim q}[\mathcal{O}(x)] + \gamma_c \mathbb{E}_{x \sim q}[\mathcal{P}(x)] \tag{23}$$

Expanding the KL divergence:

$$f_q = \mathbb{E}_{x \sim q}\left[\log \frac{q(x)}{p(x^*|\mathbf{g})}\right] + \gamma_o \mathbb{E}_{x \sim q}[\mathcal{O}(x)] + \gamma_c \mathbb{E}_{x \sim q}[\mathcal{P}(x)] \tag{24}$$

$$= \mathbb{E}_{x \sim q}\left[\log q(x) - \log p(x^*|\mathbf{g}) + \gamma_o \mathcal{O}(x) + \gamma_c \mathcal{P}(x)\right] \tag{25}$$

$$= \mathbb{E}_{x \sim q}\left[\log q(x) - \log p(x^*|\mathbf{g}) - \log \exp(-\gamma_o \mathcal{O}(x) - \gamma_c \mathcal{P}(x))\right] \tag{26}$$

$$= \mathbb{E}_{x \sim q}\left[\log q(x) - \log(p(x^*|\mathbf{g}) \exp(-\gamma_o \mathcal{O}(x) - \gamma_c \mathcal{P}(x)))\right] \tag{27}$$

$$= \mathbb{E}_{x \sim q}\left[\log q(x) - \log(Z \cdot \tilde{p}(x|\mathbf{g}))\right] \tag{28}$$

$$= \mathbb{E}_{x \sim q}\left[\log \frac{q(x)}{\tilde{p}(x|\mathbf{g})}\right] - \log Z \tag{29}$$

$$= D_{\mathrm{KL}}(q(x)\|\tilde{p}(x|\mathbf{g})) - \log Z \tag{30}$$

Therefore:

$$\boxed{\min_q f_q = \min_q D_{\mathrm{KL}}(q(x)\|\tilde{p}(x|\mathbf{g})) - \log Z} \tag{31}$$

Since $\log Z$ is constant with respect to $q$, the two optimization problems are equivalent. $\qquad\square$

## B    PROOF OF THE MORE COMPACT DISTRIBUTION

This part, we prove the 2: *Proof.* By definition, the Lagrangian-guided distribution is:

$$\tilde{p}(\mathcal{X}|\mathbf{g}) \cong p_0(\mathcal{X}|\mathbf{g}) \exp(-\gamma_o \mathcal{O}(\mathcal{X}) - \gamma_c \mathcal{P}(\mathcal{X})) \tag{32}$$

Define the guidance factor $w(\mathcal{X}) = \exp(-\gamma_o \mathcal{O}(\mathcal{X}) - \gamma_c \mathcal{P}(\mathcal{X}))$. Note that:

- $w(\mathcal{X}) \leq 1$ for all $\mathcal{X}$ (since $\mathcal{O}(\mathcal{X}) \geq 0$ and $\mathcal{P}(\mathcal{X}) \geq 0$)
- $w(\mathcal{X})$ is larger for solutions with smaller objective values and constraint violations

Let $\mathcal{X}_0^* = \arg\max_{\mathcal{X}} p_0(\mathcal{X}|\mathbf{g})$ and $\mathcal{X}_g^* = \arg\max_{\mathcal{X}} \tilde{p}(\mathcal{X}|\mathbf{g})$.

Since $\tilde{p}(\mathcal{X}|\mathbf{g}) = p_0(\mathcal{X}|\mathbf{g})w(\mathcal{X})$, we have:

$$\max_{\mathcal{X}} \tilde{p}(\mathcal{X}|\mathbf{g}) = \max_{\mathcal{X}}[p_0(\mathcal{X}|\mathbf{g})w(\mathcal{X})] \tag{33}$$

For any $\mathcal{X} \in \Omega_2$, we have:

$$\tilde{p}(\mathcal{X}|\mathbf{g}) \geq \alpha \max_{\mathcal{X}'} \tilde{p}(\mathcal{X}'|\mathbf{g}) \tag{34}$$

$$p_0(\mathcal{X}|\mathbf{g})w(\mathcal{X}) \geq \alpha \max_{\mathcal{X}'}[p_0(\mathcal{X}'|\mathbf{g})w(\mathcal{X}')] \tag{35}$$

Since $w(\mathcal{X}) \leq 1$ and $\max_{\mathcal{X}'}[p_0(\mathcal{X}'|\mathbf{g})w(\mathcal{X}')] \leq \max_{\mathcal{X}'} p_0(\mathcal{X}'|\mathbf{g})$, we have:

For $\mathcal{X} \in \Omega_2$:

$$p_0(\mathcal{X}|\mathbf{g}) \geq \frac{\alpha \max_{\mathcal{X}'}[p_0(\mathcal{X}'|\mathbf{g})w(\mathcal{X}')]}{w(\mathcal{X})} \tag{36}$$

$$\geq \alpha \max_{\mathcal{X}'}[p_0(\mathcal{X}'|\mathbf{g})w(\mathcal{X}')] \quad \text{(since } w(\mathcal{X}) \leq 1) \tag{37}$$

$$\geq \alpha\beta \max_{\mathcal{X}'} p_0(\mathcal{X}'|\mathbf{g}) \tag{38}$$

where $\beta = \min_{\mathcal{X}:\tilde{p}(\mathcal{X}|\mathbf{g}) \geq \alpha \max \tilde{p}} w(\mathcal{X}) > 0$.

Since the Lagrangian guidance preferentially increases the probability of high-quality solutions (those with low $\mathcal{O}(\mathcal{X})$ and $\mathcal{P}(\mathcal{X})$), the high-probability region $\Omega_2$ under the guided distribution is concentrated on solutions that were already relatively good under the original distribution $p_0$.

Therefore, $\Omega_2 \subseteq \Omega_1$, which means the Lagrangian guidance mechanism effectively tightens the solution space by concentrating probability mass on higher-quality solutions. $\square$

## C  PROBABILISTIC DERIVATION OF LAGRANGIAN-AWARE SDE

We derive the Lagrangian-relaxed SDE from first principles by considering the evolution of probability distributions under constraints in Theorem 4:

*Proof.* We construct the proof by establishing the reverse-time SDE for the Lagrangian-guided distribution and deriving the corresponding score function.

Define the Lagrangian-guided target distribution at time $t = 0$:

$$\tilde{p}_0(x|\mathbf{g}) \propto p_{\text{data}}(x) \exp\left(-\gamma_o \mathcal{O}(x) - \gamma_c \mathcal{P}(x)\right) \tag{39}$$

The forward diffusion process transforms $\tilde{p}_0(x|\mathbf{g})$ according to the standard forward SDE:

$$d\mathcal{X}_t = -\frac{1}{2}\beta(t)\mathcal{X}_t dt + \sqrt{\beta(t)}dW_t \tag{40}$$

with corresponding Fokker-Planck equation:

$$\frac{\partial \tilde{p}_t}{\partial t} = \frac{1}{2}\beta(t)\nabla \cdot (\mathcal{X}\tilde{p}_t + \nabla\tilde{p}_t) \tag{41}$$

By Anderson's theorem (Anderson, 1982), the reverse-time SDE that samples from $\tilde{p}_t(x|\mathbf{g})$ is:

$$d\mathcal{X}_t = \left[-\frac{1}{2}\beta(t)\mathcal{X}_t - \beta(t)\nabla_{\mathcal{X}} \log \tilde{p}_t(\mathcal{X}_t|\mathbf{g})\right] dt + \sqrt{\beta(t)}d\bar{W}_t \tag{42}$$

For the Lagrangian-guided distribution, we decompose the score function:

$$\nabla_{\mathcal{X}} \log \tilde{p}_t(\mathcal{X}_t|\mathbf{g}) = \nabla_{\mathcal{X}} \log p_t^{\text{data}}(\mathcal{X}_t) + \nabla_{\mathcal{X}} \log \exp\left(-\gamma_o \mathcal{O}(x_t) - \gamma_c \mathcal{P}(x_t)\right) \tag{43}$$

Taking the gradient of the exponential term:

$$\nabla_{\mathcal{X}} \log \exp\left(-\gamma_o \mathcal{O}(x_t) - \gamma_c \mathcal{P}(x_t)\right) = -\gamma_o \nabla_{\mathcal{X}}\mathcal{O}(x_t) - \gamma_c \nabla_{\mathcal{X}}\mathcal{P}(x_t) \tag{44}$$

where $x_t = \mathbf{Vec}(\mathcal{X}_t)$ is the vectorized form. Combining the terms, the complete score function becomes:

$$\nabla_{\mathcal{X}} \log \tilde{p}_t(\mathcal{X}_t|\mathbf{g}) = \partial_{\mathcal{X}} p(\mathcal{X}^*|\mathbf{g}) - \gamma_o \partial_x \mathcal{O}(x_t) - \gamma_c \partial_x \mathcal{P}(x_t) \tag{45}$$

Substituting equation equation 45 into the reverse SDE equation 42:

$$d\mathcal{X}_t = \left[-\frac{1}{2}\beta(t)\mathcal{X}_t - \beta(t)\left(\partial_{\mathcal{X}} p(\mathcal{X}^*|\mathbf{g}) - \gamma_o \partial_x \mathcal{O}(x_t) - \gamma_c \partial_x \mathcal{P}(x_t)\right)\right] dt + \sqrt{\beta(t)}d\bar{W}_t$$

The overall score function used in both training and sampling is therefore:

$$s_\theta^*(\mathcal{X}_t, t, \mathbf{g}) = \partial_{\mathcal{X}} p(\mathcal{X}^*|\mathbf{g}) - \gamma_o \partial_x \mathcal{O}(x_t) - \gamma_c \partial_x \mathcal{P}(x_t) \tag{46}$$

This modified score function integrates the data distribution score with optimization-aware gradient signals from both the objective function $\mathcal{O}(x_t)$ and penalty function $\mathcal{P}(x_t)$, effectively guiding the diffusion trajectory toward feasible and optimal regions during the sampling process. $\square$

# D   PROOF OF THE APPROXIMATION FOR VRG

To prove the approximation bound,we firstly introduce the two approximation errors assumptions:

**Assumption D.1** (Score Network Approximation). *The trained score network $s_\theta(\mathcal{X}_t, t, \mathbf{g})$ approximates the target score $s_\theta^*(\mathcal{X}_t, t, \mathbf{g})$ with bounded error:*

$$\|s_\theta(\mathcal{X}_t, t, \mathbf{g}) - s_\theta^*(\mathcal{X}_t, t, \mathbf{g})\|_2 \le \delta, \quad \forall t \in [0, T] \tag{47}$$

**Assumption D.2** (Bounded Domain). *The solution domain is bounded: $\|x\|_2 \le R$ for some $R > 0$.*

## D.1   PROOF OF THE VISUAL-GUIDED OPTIMALITY GAP BOUND

Then we introduce following lemmas to prepare the error bound for the VRG different parts:

**Lemma 1** (Visual Space Isometry). *For any two solutions $x, x' \in \mathbb{R}^n$ and their visual representations $\mathcal{X} = \mathbf{Vec}^{-1}(x)$, $\mathcal{X}' = \mathbf{Vec}^{-1}(x')$:*

$$\|x - x'\|_2 = \|\mathcal{X} - \mathcal{X}'\|_F \tag{48}$$

*Proof.* By definition of the Vec transformation, for $\mathcal{X} \in \mathbb{R}^{h \times w}$:

$$[\mathbf{Vec}(\mathcal{X})]_k = \mathcal{X}_{i,j} \quad \text{where } k = (i-1)w + j \tag{49}$$

This is a bijective reshaping that preserves all elements. Therefore:

$$\|x - x'\|_2^2 = \sum_{k=1}^n (x_k - x_k')^2 \tag{50}$$

$$= \sum_{i=1}^h \sum_{j=1}^w (\mathcal{X}_{i,j} - \mathcal{X}_{i,j}')^2 \tag{51}$$

$$= \|\mathcal{X} - \mathcal{X}'\|_F^2 \tag{52}$$

$\square$

**Lemma 2.** *Let $\{\mathcal{X}_t\}_{t=T}^0$ be the trajectory generated by the reverse SDE:*

$$d\mathcal{X}_t = \left[-\frac{1}{2}\beta(t)\mathcal{X}_t - \beta(t)s_\theta(\mathcal{X}_t, t, \mathbf{g})\right] dt + \sqrt{\beta(t)}d\bar{W}_t \tag{53}$$

*and let $\{\mathcal{X}_t^*\}_{t=T}^0$ be the trajectory with the true score $s_\theta^*$. Then:*

$$\mathbb{E}\left[\|\mathcal{X}_0 - \mathcal{X}_0^*\|_F^2\right] \le \delta^2 \cdot \sum_{t=1}^T \beta_t \tag{54}$$

*Proof.* Define the error process $e_t = \mathcal{X}_t - \mathcal{X}_t^*$. The dynamics of $e_t$ satisfy:

$$de_t = \left[-\frac{1}{2}\beta(t)e_t - \beta(t)(s_\theta(\mathcal{X}_t, t, \mathbf{g}) - s_\theta^*(\mathcal{X}_t^*, t, \mathbf{g}))\right] dt \tag{55}$$

Note that both processes share the same Brownian motion, so it cancels in the difference.

Taking the squared norm and applying Itô's lemma:

$$d\|e_t\|_F^2 = 2\langle e_t, de_t \rangle \tag{56}$$

$$= 2\langle e_t, -\frac{1}{2}\beta(t)e_t - \beta(t)(s_\theta - s_\theta^*)\rangle dt \tag{57}$$

$$= -\beta(t)\|e_t\|_F^2 dt - 2\beta(t)\langle e_t, s_\theta - s_\theta^*\rangle dt \tag{58}$$

Using Cauchy-Schwarz and the score error bound (Assumption D.1):

$$|\langle e_t, s_\theta - s_\theta^*\rangle| \le \|e_t\|_F \cdot \|s_\theta - s_\theta^*\|_2 \le \|e_t\|_F \cdot \delta \tag{59}$$

Applying Young's inequality $2ab \leq a^2 + b^2$:

$$-2\beta(t)\langle e_t, s_\theta - s_\theta^* \rangle \leq \beta(t)\|e_t\|_F^2 + \beta(t)\delta^2 \tag{60}$$

Therefore:

$$d\|e_t\|_F^2 \leq -\beta(t)\|e_t\|_F^2 dt + \beta(t)\|e_t\|_F^2 dt + \beta(t)\delta^2 dt = \beta(t)\delta^2 dt \tag{61}$$

Integrating from $t = T$ to $t = 0$ (with $e_T = 0$ since both start from the same initial distribution):

$$\mathbb{E}[\|e_0\|_F^2] = \mathbb{E}[\|\mathcal{X}_0 - \mathcal{X}_0^*\|_F^2] \leq \delta^2 \sum_{t=1}^{T} \beta_t \tag{62}$$

$\square$

**Lemma 3** (Guided Distribution Concentration). *For $\mathcal{X}_0^*$ sampled from the true guided distribution $\tilde{p}(\mathcal{X}|\mathbf{g})$:*

$$\mathbb{E}_{\tilde{p}}\left[\mathcal{O}(\textbf{Vec}(\mathcal{X}_0^*))\right] \leq \frac{1}{\gamma_o}\left(D_{KL}(p\|\tilde{p}) + \log Z\right) \tag{63}$$

*and similarly for the constraint penalty.*

*Proof.* From the definition of the guided distribution:

$$\tilde{p}(\mathcal{X}|\mathbf{g}) = \frac{1}{Z}p(\mathcal{X}^*|\mathbf{g})\exp(-\gamma_o\mathcal{O}(\textbf{Vec}(\mathcal{X})) - \gamma_c\mathcal{P}(\textbf{Vec}(\mathcal{X}))) \tag{64}$$

Taking the logarithm:

$$\log\tilde{p}(\mathcal{X}|\mathbf{g}) = \log p(\mathcal{X}^*|\mathbf{g}) - \gamma_o\mathcal{O}(\textbf{Vec}(\mathcal{X})) - \gamma_c\mathcal{P}(\textbf{Vec}(\mathcal{X})) - \log Z \tag{65}$$

Rearranging for $\mathcal{O}$:

$$\gamma_o\mathcal{O}(\textbf{Vec}(\mathcal{X})) = \log p(\mathcal{X}^*|\mathbf{g}) - \log\tilde{p}(\mathcal{X}|\mathbf{g}) - \gamma_c\mathcal{P}(\textbf{Vec}(\mathcal{X})) - \log Z \tag{66}$$

Taking expectation under $\tilde{p}$:

$$\gamma_o\mathbb{E}_{\tilde{p}}[\mathcal{O}(\textbf{Vec}(\mathcal{X}))] = \mathbb{E}_{\tilde{p}}[\log p - \log\tilde{p}] - \gamma_c\mathbb{E}_{\tilde{p}}[\mathcal{P}] - \log Z \tag{67}$$
$$= -D_{KL}(\tilde{p}\|p) - \gamma_c\mathbb{E}_{\tilde{p}}[\mathcal{P}] - \log Z \tag{68}$$

Since $D_{KL}(\tilde{p}\|p) \geq 0$ and $\mathbb{E}_{\tilde{p}}[\mathcal{P}] \geq 0$:

$$\mathbb{E}_{\tilde{p}}[\mathcal{O}(\textbf{Vec}(\mathcal{X}))] \leq \frac{-\log Z}{\gamma_o} \tag{69}$$

For the normalization constant, note that $Z \leq 1$ (since we're multiplying a probability density by an exponential of negative terms), so $-\log Z \geq 0$, and the bound is informative when $Z$ is not too small. $\square$

**Lemma 4** (Lagrangian Duality Gap). *For any solution $\tilde{x}$ and optimal Lagrange multiplier $\lambda^* \geq 0$:*

$$c^\top\tilde{x} - c^\top x^* \leq (\lambda^*)^\top(A\tilde{x} - b)^+ + (z_{MILP}^* - z_{Lagrangian}^*) \tag{70}$$

*where $(v)^+ = \max\{v, 0\}$ element-wise.*

*Proof.* From Lagrangian duality theory, for the optimal $\lambda^*$:

$$z_{\text{Lagrangian}}^* = \min_x\{c^\top x + (\lambda^*)^\top(Ax - b)\} \leq z_{\text{MILP}}^* \tag{71}$$

For any $\tilde{x}$:

$$c^\top\tilde{x} + (\lambda^*)^\top(A\tilde{x} - b) \geq z_{\text{Lagrangian}}^* \tag{72}$$

Rearranging:

$$c^\top\tilde{x} \geq z_{\text{Lagrangian}}^* - (\lambda^*)^\top(A\tilde{x} - b) \tag{73}$$

For the upper bound, when $A\tilde{x} - b \leq 0$ (feasible), we have $c^\top \tilde{x} \geq z_{\text{MILP}}^*$ (optimality). When constraints are violated:

$$c^\top \tilde{x} - c^\top x^* = c^\top \tilde{x} - z_{\text{MILP}}^* \tag{74}$$

$$\leq c^\top \tilde{x} - z_{\text{Lagrangian}}^* \tag{75}$$

$$\leq (\lambda^*)^\top (A\tilde{x} - b)^+ + (z_{\text{MILP}}^* - z_{\text{Lagrangian}}^*) \tag{76}$$

For violated constraints $(A\tilde{x} - b)_i > 0$, the Lagrangian penalty accounts for them, while for satisfied constraints the penalty is zero. $\qquad\square$

**Lemma 5** (Sampling Concentration). *With $T$ diffusion steps and bounded domain $\|x\|_2 \leq R$, for any $\epsilon > 0$:*

$$\mathbb{P}\left( \|\tilde{x} - \mathbb{E}[\tilde{x}]\|_2 > R\sqrt{\frac{2\log(2/\epsilon)}{T}} \right) \leq \epsilon \tag{77}$$

*Proof.* This follows from standard concentration bounds for Langevin dynamics and score-based sampling. The discretization of the reverse SDE introduces a sampling error that decreases as $\mathcal{O}(1/\sqrt{T})$.

Specifically, using the bounded domain assumption and Hoeffding's inequality applied to the discretized diffusion process, each coordinate of $\tilde{x}$ has sub-Gaussian tails with parameter $\sigma^2 = R^2/T$.

By standard sub-Gaussian concentration:

$$\mathbb{P}(|\tilde{x}_i - \mathbb{E}[\tilde{x}_i]| > t) \leq 2\exp\left( -\frac{t^2 T}{2R^2} \right) \tag{78}$$

Taking a union bound over $n$ coordinates and setting $t = R\sqrt{\frac{2\log(2n/\epsilon)}{T}}$:

$$\mathbb{P}\left( \|\tilde{x} - \mathbb{E}[\tilde{x}]\|_2 > R\sqrt{\frac{2n\log(2n/\epsilon)}{T}} \right) \leq \epsilon \tag{79}$$

$\qquad\square$

**Theorem 5** (Visual-Guided Optimality Gap Bound). *Let $x^*$ be the optimal solution with $z_{MILP}^* = c^\top x^*$, and let $\tilde{x}$ be sampled from the Lagrangian-guided distribution $\tilde{p}(\mathcal{X}|\mathbf{g})$. Under Assumptions D.1 and D.2, with probability at least $1 - \epsilon$:*

$$|c^\top \tilde{x} - c^\top x^*| \leq \frac{\|c\|_2}{\gamma_o} \cdot \mathcal{E}_{score} + \frac{\|c\|_2 \cdot \|\lambda^*\|_2 \|A\|_F}{\gamma_o \gamma_c} \cdot \mathcal{P}(\tilde{x}) + \|c\|_2 \sqrt{\frac{2R^2 \log(2/\epsilon)}{T}} \tag{80}$$

*where $\mathcal{E}_{score} = \delta \cdot \sqrt{\sum_{t=1}^{T} \beta_t}$ and $\lambda^*$ is the optimal Lagrange multiplier.*

*Proof.* We clarify the three intermediate reference points to decompose the error:

- $x^*$: The true optimal solution of the MILP

- $x_{\text{guided}}^*$: The mode of the ideal guided distribution $\tilde{p}(\mathcal{X}|\mathbf{g})$ with perfect score

- $\hat{x}$: The expected solution when sampling with the learned score $s_\theta$ (i.e., $\hat{x} = \mathbb{E}_{s_\theta}[\tilde{x}]$)

- $\tilde{x}$: The actual sampled solution

So for the $|c^\top \tilde{x} - c^\top x^*|$, by Cauchy-Schwarz Inequality (Schwarz, 1885), we have:

$$|c^\top \tilde{x} - c^\top x^*| = |c^\top (\tilde{x} - x^*)| \leq \|c\|_2 \|\tilde{x} - x^*\|_2 \tag{81}$$

Then, we decompose $\|\tilde{x} - x^*\|_2$ using triangle inequality with the intermediate points:

$$\boxed{\|\tilde{x} - x^*\|_2 \leq \|\tilde{x} - \hat{x}\|_2 + \|\hat{x} - x^*_{\text{guided}}\|_2 + \|x^*_{\text{guided}} - x^*\|_2} \tag{82}$$

Then, we prove the decompostion part of the Eq 82 is bounded with lemmas proved before:

From Lemma 5, the sampling process with $T$ diffusion steps satisfies:

$$\mathbb{P}\left(\|\tilde{x} - \hat{x}\|_2 > R\sqrt{\frac{2\log(2/\epsilon)}{T}}\right) \leq \epsilon \tag{83}$$

Therefore, with probability at least $1 - \epsilon$:

$$\|\tilde{x} - \hat{x}\|_2 \leq R\sqrt{\frac{2\log(2/\epsilon)}{T}} \tag{84}$$

Multiplying by $\|c\|_2$:

$$\|c\|_2\|\tilde{x} - \hat{x}\|_2 \leq \|c\|_2\sqrt{\frac{2R^2\log(2/\epsilon)}{T}} \tag{85}$$

By Lemma 1, we work in visual space:

$$\|\hat{x} - x^*_{\text{guided}}\|_2 = \|\hat{\mathcal{X}} - \mathcal{X}^*_{\text{guided}}\|_F \tag{86}$$

From Lemma 2, the SDE trajectory error due to score approximation:

$$\mathbb{E}[\|\hat{\mathcal{X}} - \mathcal{X}^*_{\text{guided}}\|_F^2] \leq \delta^2 \sum_{t=1}^{T} \beta_t \tag{87}$$

By Jensen's inequality (Jensen, 1906):

$$\mathbb{E}[\|\hat{\mathcal{X}} - \mathcal{X}^*_{\text{guided}}\|_F] \leq \sqrt{\mathbb{E}[\|\hat{\mathcal{X}} - \mathcal{X}^*_{\text{guided}}\|_F^2]} \leq \delta\sqrt{\sum_{t=1}^{T} \beta_t} = \mathcal{E}_{\text{score}} \tag{88}$$

The guidance strength $\gamma_o$ appears because the score function is:

$$s^*_\theta = \nabla_{\mathcal{X}} \log p(\mathcal{X}^*|\mathbf{g}) - \gamma_o \nabla_x \mathcal{O}(x) - \gamma_c \nabla_x \mathcal{P}(x) \tag{89}$$

The effective error in the objective direction is scaled by $1/\gamma_o$ since larger $\gamma_o$ amplifies the correction signal. Thus:

$$\|c\|_2\|\hat{x} - x^*_{\text{guided}}\|_2 \leq \frac{\|c\|_2}{\gamma_o} \cdot \mathcal{E}_{\text{score}} \tag{90}$$

From Lemma 4 (Lagrangian duality):

$$c^\top x^*_{\text{guided}} - c^\top x^* \leq (\lambda^*)^\top (A x^*_{\text{guided}} - b)^+ \tag{91}$$

For the sampled solution $\tilde{x}$, the constraint violation contributes:

$$(\lambda^*)^\top (A\tilde{x} - b)^+ \leq \|\lambda^*\|_2\|(A\tilde{x} - b)^+\|_2 \quad \text{(Cauchy-Schwarz)} \tag{92}$$

The constraint penalty is defined as $\mathcal{P}(\tilde{x}) = \|(A\tilde{x} - b)^+\|_1$. Using norm equivalence:

$$\|(A\tilde{x} - b)^+\|_2 \leq \|(A\tilde{x} - b)^+\|_1 = \mathcal{P}(\tilde{x}) \tag{93}$$

The guidance weights $\gamma_o, \gamma_c$ control how strongly the distribution penalizes objective deviation and constraint violation. The effective contribution to the objective gap is:

$$\|c\|_2\|x^*_{\text{guided}} - x^*\|_2 \leq \frac{\|c\|_2 \cdot \|\lambda^*\|_2\|A\|_F}{\gamma_o\gamma_c} \cdot \mathcal{P}(\tilde{x}) \tag{94}$$

r

Table 9: Problem sizes for each Benchmarks at different scales (varibles,constraints)

|        | SC           | MIS          | CA           | FC           |
|--------|--------------|--------------|--------------|--------------|
| Small  | (500,400)    | (500,2998)   | (300,100)    | (400,499)    |
| Middle | (500,1000)   | (1000,2000)  | (500,1000)   | (1000,1249)  |
| Large  | (1000,1500)  | (1000,3993)  | (1200,4000)  | (1500,5200)  |

Finally, we have bounded each component in the decomposition equation 82:

$$|c^\top \tilde{x} - c^\top x^*| \leq \|c\|_2 \|\tilde{x} - x^*\|_2 \tag{95}$$

$$\leq \|c\|_2 \left( \|\tilde{x} - \hat{x}\|_2 + \|\hat{x} - x^*_{\text{guided}}\|_2 + \|x^*_{\text{guided}} - x^*\|_2 \right) \tag{96}$$

$$\leq \frac{\|c\|_2}{\gamma_o} \cdot \mathcal{E}_{\text{score}} + \frac{\|c\|_2 \cdot \|\lambda^*\|_2 \|A\|_F}{\gamma_o \gamma_c} \cdot \mathcal{P}(\tilde{x}) + \|c\|_2 \sqrt{\frac{2R^2 \log(2/\epsilon)}{T}} \tag{97}$$

$$\square$$

# E    DETAILS OF EXPERIMENT SETTINGS

We clarify the detailed settings of Experiments.

## E.1    SIZE OF THE BENCHMARKS

We firstly give the deterministic for all instances.

## E.2    $\Delta$ IN THE SEARCHING

Table 10: The value of the raidus $\Delta$ in Small, Middle, Large Public Benchmarks.

|                      | SC   | MIS  | CA   | FC   |
|----------------------|------|------|------|------|
| $\Delta_{\text{Small}}$  | 500  | 400  | 1000 | 500  |
| $\Delta_{\text{Middle}}$ | 1000 | 800  | 1200 | 1000 |
| $\Delta_{\text{Large}}$  | 1500 | 1500 | 1500 | 1500 |

## E.3    PARAMETER IN **VRG**

For small-size benchamarks,we sample for 10 steps and provide 3 samples to get the multi-modality of the visual relaxed score function; For IS and SC medium-scale benchmarks,we sample 10 steps for 8 samples,others are 5 steps for 10 samples; For Large-scale benchmarks, we sample 8 steps for 5 samples.

## E.4    PARAMETER IN PREDICT-AND-SEARCH BASED BASELINES

For predict and search baseline, we use the $k_1, k_2 = 20, 30$, and the same value of the radius with the VRG.For all methods, the training dataset are 200 instances.

## E.5    RESOLUTION SIZE

We provide the detailed resolution size for three-size public benchmarks in Table 11 for our **VRG** generation methods.

Table 11: The value resolution size $(h, w)$ in Small, Middle, Large Public Benchmarks.

|        | SC      | MIS     | CA      | FC      |
|--------|---------|---------|---------|---------|
| Small  | (20,25) | (20,25) | (15,20) | (20,20) |
| Middle | (20,25) | (25,40) | (25,20) | (25,40) |
| Large  | (25,40) | (25,40) | (40,30) | (30,50) |

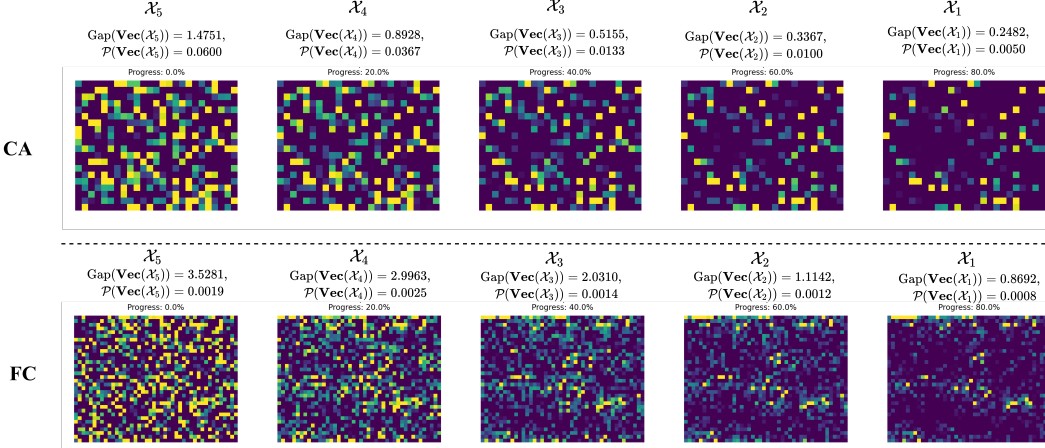

Figure 3: Qualitative illustration of how visual relaxed generation schedules affect the generation quality of continuous diffusion for 5-step sampling process visualization for the Facility location(FC) with the medium scale. $\text{Gap}(\mathbf{Vec}(\mathcal{X}_t))) = |\mathcal{O}(\mathbf{Vec}(\mathcal{X}_t))|/\mathbf{BKS}$,the lower the better.

## F    QUALITATIVE ILLUSTRATION

### F.1    GENERATION PROCESS.

We give the generation process visualization on the medium CA and FA benchmarks, which illustrate that:with the sampling going on, the image corresponded solution can be guided to the optimal feasible region with the lower and lower $\mathcal{O}(\mathbf{Vec}(\mathcal{X}_t))$ and $\mathcal{P}(\mathbf{Vec}(\mathcal{X}_t))$.

### F.2    MULTI MODAL.

We visulize the multi-modality of the generation preocess.The step $t1$ means the sampling step, which indicates that $\mathcal{X}_t = \mathcal{X}_{t_{all}-t1}$,where the $t_{all}$ means the whole steps of the sampling.

## G    PARAMETER INFLUENCE ON GENERATION PROGRESS.

In this section, we analyze the role of each hyperparameter in our training and generation process.

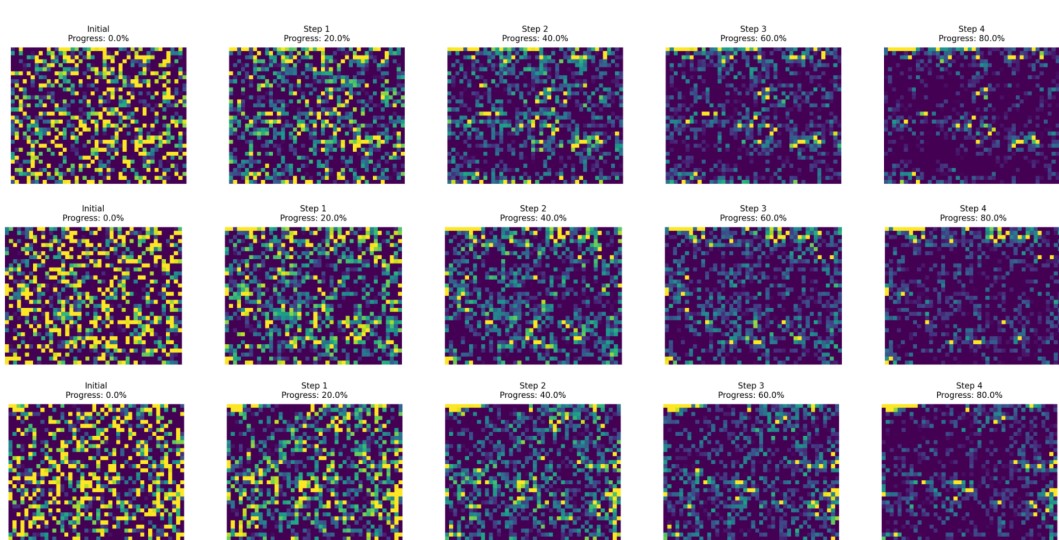

Figure 4: Multi-modality sampling on Large scale FC problem. Through Lagrangian-guided multimodal sampling, we can discover that the images corresponding to the final sampled solutions exhibit different morphologies, which reflects the multimodal nature based on Lagrangian scores and improves the robustness of subsequently constructed search regions.

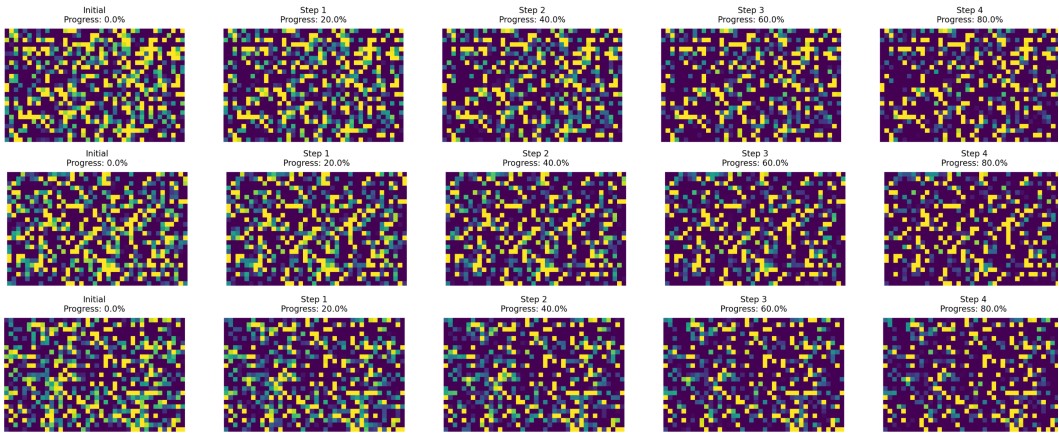

Figure 5: Multi-modality sampling on Large-scale MIS problem.

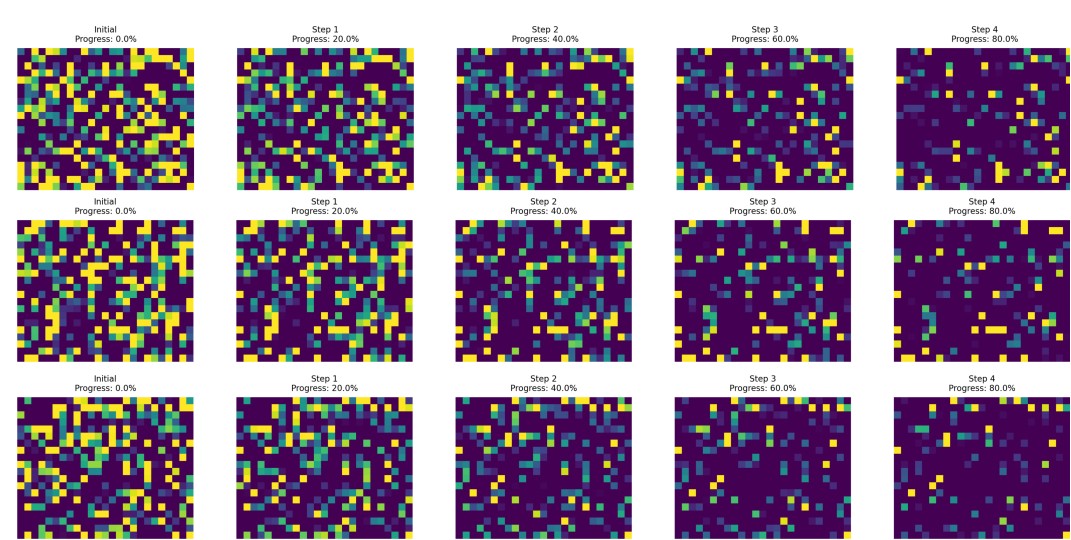

Figure 6: Multi-modality sampling on Medium-scale CA problem.

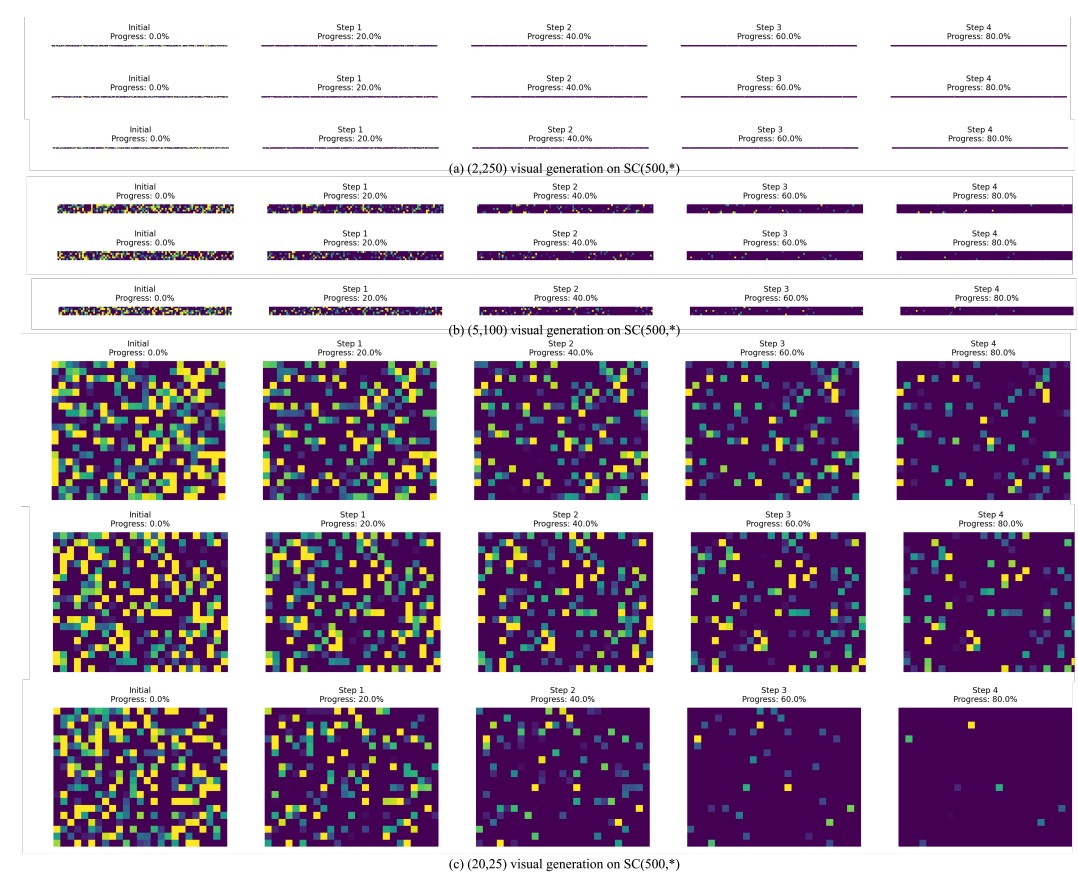

Figure 7: Different $(h, w)$ generation for 5 progress generations.

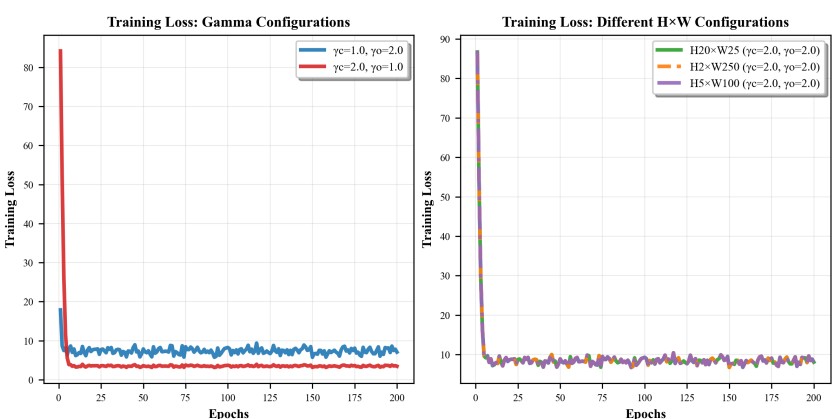

Figure 8: Different $(h, w)$ and $\gamma_o, \gamma_c$ influence on the approximation loss for target score.

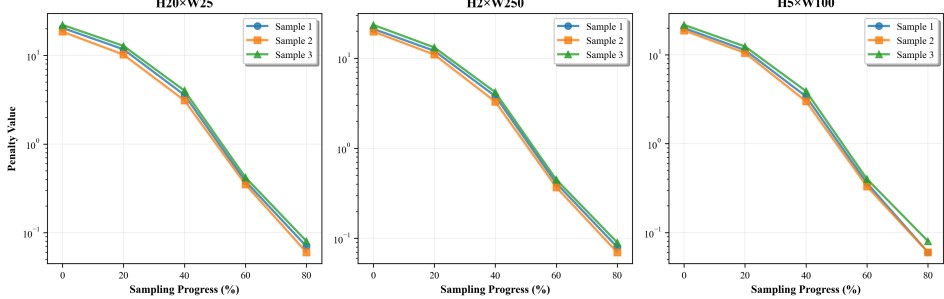

Figure 9: Penalty value trajectories during the iterative generation process for different $(h, w)$ configurations. The decreasing penalty values demonstrate that the generated solutions progressively satisfy the linear constraints throughout the sampling process.