# OpenReview forum: "Lagrangian Meets Diffusion: Feasibility-aware Generative Modeling for Mixed Integer Linear Programming"
_ICLR.cc/2026/Conference — Submitted to ICLR 2026_

### Official Review · Reviewer_HQwA · 2025-10-24

**Soundness:** 2
**Presentation:** 2
**Contribution:** 2
**Rating:** 2
**Confidence:** 5

**Summary:**

This paper proposes VRG, a generative framework that integrates diffusion models with a Lagrangian-guided optimization mechanism for solving MILPs. The authors systematically revisit the limitations of existing Predict-and-Search frameworks and introduce a novel representation method for solution vectors. Specifically, they employ a bijective visual transformation that reshapes each solution vector into an image, where each variable corresponds to a pixel, enabling convolutional diffusion networks to capture interdependencies among variables. Furthermore, to improve the quality and feasibility of generated solutions, the framework incorporates objective optimality and constraint violation penalties as regularization terms within the diffusion process.

**Strengths:**

1. This paper introduces a Lagrangian-guided diffusion process where both the objective optimality and constraint violation are used as guidance terms, effectively steering the generation toward feasible and high-quality solutions.
2. Across four standard MILP benchmarks, VRG consistently outperforms PaS and CoPaS in both solution quality and solver efficiency, while completely avoiding infeasible subproblems.

**Weaknesses:**

1. To capture interdependencies among variables, the authors reshape the solution vector into an image via a bijective transformation. However, this transformation is essentially a linear index mapping that may distort the inherent structural relationships among variables. Variables that are tightly coupled within the same constraint may end up being placed far apart in the image space, while unrelated variables may become adjacent. In contrast, the bipartite graph representation naturally preserves the structural relations between variables and constraints, making it a more faithful representation for MILP problems.
2. Although the authors emphasize Lagrangian relaxation, its core advantage of providing dual bounds that can accelerate convergence is not fully exploited here. The proposed approach only incorporates the optimality gap and constraint violations as guidance terms in the diffusion process, which captures the conceptual idea but not the computational strength of classical Lagrangian dual methods.
3. The experimental evaluation lacks comprehensive ablation studies to isolate the contribution of each component. Without such analysis, it is difficult to assess the true source of the performance gains and the reliability of the reported results.

**Questions:**

1. Could you elaborate on the design rationale of the visual–vector transformation and explain, with evidence, why this image-based representation can capture interdependencies among variables better than the bipartite graph representation?
2. During the visual–vector transformation, the choices of $h$ and $w$ affect how closely related variables are placed in the image space. Could you provide a sensitivity analysis showing how different ($h$, $w$) settings impact performance and the preservation of variable dependencies?
3. Since $\gamma_o$ and $\gamma_c$ control the strength of optimality and feasibility guidance in the diffusion process, please include an extensive analysis of these hyperparameters. In particular, report their effect on solution quality and feasibility across scales and benchmarks, and provide practical guidelines for setting them.
4. To show the benefit of the guidance mechanism, please report results for the diffusion model without the guidance terms. In addition, diffusion models often require deeper architectures than GNN-based solvers. Could you include a study of performance as a function of network depth?
5. Could you add some new baselins? such as [1] and [2].

[1] APOLLO-MILP: An Alternating Prediction–Correction Neural Solving Framework for MILP. ICLR 2025.

[2] Effective Generation of Feasible Solutions for Integer Programming via Guided Diffusion. KDD 2024.

---

> ### Author Response · Authors · 2025-12-04
>
> Thanks for your insightful comments. We have revised the paper accordingly, with all changes highlighted in orange.
>
> **Reply to Weakness 1 and Questions 1 & 4**
>
> Thank you for this insightful question. We sincerely appreciate the opportunity to clarify our design rationale and the role of visual representation in VRG.
>
> **1. We do leverage the bipartite graph representation.**
> As described in the paper, the MILP problem is first encoded into a graph embedding $g$ using a GNN, which serves as the conditional control for our visual generative model. Thus, our model acknowledges and utilizes the structural representation capability of the bipartite graph.
>
> **2. The limitation we address is in the generation/prediction phase.**
> Existing end-to-end predict-and-search methods typically use graph embeddings to directly predict the probability of each variable dimension independently. They may lose the ability to model interdependencies among variables during generation (prediction), as each variable's prediction is made in isolation.
>
> **3. Why the image-based representation helps.**
> By reshaping the solution vector into an image and using a U-Net architecture for generation, we leverage:
> - **Receptive fields**: Convolutional layers capture both local and global dependencies among variables.
> - **Iterative refinement**: The denoising process allows the model to progressively refine variable assignments while considering their interactions.
>
> In summary, our core idea of reshaping the solution vector into a visual image does not aim to replace the structural representation of the MILP with visual data. Instead, we employ a graph neural network (GNN) to embed the structural information of the MILP, which then serves as a condition for generating the MILP solution. This concept of utilizing an image for generation finds a parallel in the approach of DeepSeek-OCR [1], which maps sequential dependencies in text into spatial and contextual dependencies within an image. We posit that our method demonstrates strong scalability and practical value. Experimental results further validate the superiority of our approach in generating high-quality solutions.
>
> > [1] DeepSeek AI. DeepSeek-OCR: Contexts Optical Compression, 2025.
>
> **4. Empirical evidence on benefits from visual representations.**
> Our experimental results demonstrate that, by using end-to-end generation without hyperparameter tuning, our method outperforms approaches that directly use GCN as a solution predictor, which typically require carefully tuned parameters such as $k_1$ and $k_2$ to define the trust region. Moreover, our method consistently maintains feasibility across all problem instances while reducing the number of nodes explored by Gurobi and SCIP. To further validate the effectiveness of the visual representation, we conducted an ablation study by replacing the U-Net with an MLP that directly predicts the solution vector. Table 8 shows the approximation loss for the solution vector generation.
>
> **5. Empirical evidence on the number of visual encoder layers for score estimation.**
> Following your suggestions, we conducted an ablation study on the depth of the U-Net-based generation encoder and its effect on the approximations of the target score (Table 8 in the revised paper). The results show that the number of encoder layers has negligible impact on the approximation error for the guided score for the same $\gamma_o, \gamma_c$. This suggests VRG is robust to architectural depth and highlights the potential scalability of our solution-generation framework.
>
> **Table 8: Ablation study on visual representation and backbone layers (l) with $(\gamma_o, \gamma_c) = (1, 1)$, $(h, w) = (20, 25)$. Results are averaged over 40 instances.**
>
> | Method | Backbone | SC(500,800) Error (↓) |
> |--------|----------|----------------------|
> | VRG w/o Visual | MLP | > 1.2588 × 10⁷ |
> | VRG (Ours) | U-Net, l=2 | 1.9107 |
> | VRG (Ours) | U-Net, l=3 | 1.9096 |
> | VRG (Ours) | U-Net, l=4 | 1.9106 |
>
> ---

---

> ### Author Response · Authors · 2025-12-04
>
> **Reply to Weakness 2**
>
> Thank you for this insightful comment. We would like to clarify the role and rationale of Lagrangian relaxation in our method. We acknowledge that classical Lagrangian dual methods offer tight dual bounds through iterative subproblem solving and multiplier updates, which can accelerate convergence in traditional branch-and-bound frameworks. However, this procedure is computationally expensive, particularly for large-scale MILP instances, as it requires solving numerous subproblems iteratively to obtain and refine the dual bounds.
>
> **Our method takes a fundamentally different approach.** Rather than using Lagrangian relaxation to compute dual bounds directly, we leverage it as a **guidance mechanism** within a generative framework. The key insight is that we do not aim to solve the MILP through classical dual bound tightening. Instead, our goal is to efficiently generate multiple high-quality relaxed solutions that can be used to construct compact and trustworthy sub-problems, thereby reducing the search space for downstream solvers.
>
> Specifically, by relaxing the integer constraints and incorporating the optimization direction of the relaxed problem into the diffusion process, we guide the model to produce solutions that progressively approach the theoretical optimum. These high-quality relaxed solutions then define a trust region that effectively prunes the search space. The Lagrangian-inspired regularization has been integrated into our SDE formulation (Eq. 11).
>
> In the revised paper, we additionally provide new theoretical guarantees (Theorem 5 with detailed proof in Appendix D) showing that the generated solutions can approximate the optimal solution with a provable bound. This enables our method to construct high-quality sub-problems **without tuning hyperparameters**, such as $k_1$ and $k_2$ (typically required in traditional variable-fixing methods), while achieving both acceleration and improved solution quality.
>
> In summary, our method offers a complementary, data-driven alternative: by utilizing Lagrangian-inspired guidance, we efficiently identify promising solution regions without incurring the computational overhead associated with explicit dual bound computation. We believe this represents a practical trade-off between theoretical tightness and computational efficiency for large-scale MILP instances.
>
> ---

---

> ### Author Response · Authors · 2025-12-04
>
> **Reply to Weakness 3 and Questions 2, 3, 5**
>
> Thank you for your careful observation and constructive suggestions. We have conducted the requested comprehensive ablation studies. The results are presented in Table 7 and Figure 7-9 in the revised paper, with all changes highlighted in orange.
>
> **Table 7: Comparison with other ML-based effective solvers using VRG models with different $\gamma_o, \gamma_c$ and $(h, w)$. Results are averaged over 50 test instances, each solved within 100s time limit. Step denotes the number of training steps.**
>
> | Method | (h, w) | (γ_o, γ_c) | Step | Nodes (↓) | Obj (↓) |
> |--------|--------|------------|------|-----------|---------|
> | Gurobi | -- | -- | -- | 5841.46 ± 5191.29 | 28.9800 ± 3.0270 |
> | SCIP | -- | -- | -- | 3336.16 ± 3846.81 | 28.8936 ± 3.0305 |
> | PaS + SCIP | -- | -- | -- | 4873.59 ± 5177.36 | 28.9800 ± 3.0117 |
> | DiffILO | -- | -- | -- | 6252.90 ± 7095.00 | 28.9800 ± 3.0270 |
> | Apollo MILP | -- | -- | -- | 7707.42 ± 7807.04 | 28.9800 ± 3.0270 |
> | VRG + SCIP | (5, 100) | (2, 2) | 20 | 3020.22 ± 3255.73 | 28.8478 ± 3.0475 |
> | VRG + SCIP | (2, 250) | (2, 2) | 20 | 2994.58 ± 3154.28 | **28.7778 ± 3.0442** |
> | VRG + SCIP | (20, 25) | (2, 2) | 20 | 3309.38 ± 3770.38 | 28.9574 ± 3.1065 |
> | VRG + Gurobi | (5, 100) | (2, 2) | 20 | 5824.98 ± 5635.45 | 28.9800 ± 3.0270 |
> | VRG + Gurobi | (2, 250) | (2, 2) | 20 | 5824.98 ± 5635.45 | 28.9800 ± 3.0270 |
> | VRG + Gurobi | (20, 25) | (2, 2) | 20 | 5824.98 ± 5635.45 | 28.9800 ± 3.0270 |
> | VRG + SCIP | (20, 25) | (2, 1) | 20 | 2919.70 ± 3319.68 | 28.9574 ± 3.1064 |
> | VRG + SCIP | (20, 25) | (1, 1) | 20 | **2802.10 ± 2843.02** | 28.7778 ± 3.0443 |
> | VRG + SCIP | (20, 25) | (1, 2) | 20 | 2850.00 ± 2789.88 | 28.8913 ± 3.1072 |
> | VRG + Gurobi | (20, 25) | (2, 1) | 20 | 5824.98 ± 5635.44 | 28.9800 ± 3.0270 |
> | VRG + Gurobi | (20, 25) | (1, 1) | 20 | 5824.98 ± 5635.44 | 28.9800 ± 3.0270 |
> | VRG + Gurobi | (20, 25) | (1, 2) | 20 | 5824.98 ± 5635.44 | 28.9800 ± 3.0270 |
> | VRG + Gurobi | (20, 25) | (3, 4) | 20 | 5824.98 ± 5635.44 | 28.9800 ± 3.0270 |
> | VRG + Gurobi | (20, 25) | (3, 4) | 10 | 5824.98 ± 5635.44 | 28.9800 ± 3.0270 |
> | VRG + Gurobi | (20, 25) | (3, 4) | 5 | 5824.98 ± 5635.44 | 28.9800 ± 3.0270 |
> | VRG + SCIP | (20, 25) | (3, 4) | 20 | **2978.88 ± 3074.71** | 28.8913 ± 3.1071 |
> | VRG + SCIP | (20, 25) | (3, 4) | 10 | 3015.66 ± 3161.26 | 28.8913 ± 3.1071 |
> | VRG + SCIP | (20, 25) | (3, 4) | 5 | 3004.66 ± 3130.51 | 28.8913 ± 3.1071 |
>
> **1. Sensitivity analysis on visual dimensions $(h, w)$.**
> We evaluated different choices of $(h, w)$ for the visual-vector transformation. As shown in Table 7, the performance remains relatively stable across a reasonable range of $(h, w)$ settings, suggesting that our method is not overly sensitive to the choice of image dimensions within a practical range. This robustness can be attributed to the U-Net architecture's multi-scale receptive fields, which enable it to capture variable dependencies regardless of their exact spatial arrangement. Furthermore, across all tested visual dimensions, the generation process consistently denoises Gaussian noise to produce near-binary spatial visual images (with values close to 0 or 1), approximating the optimal solution.
>
> **2. Sensitivity analysis on guidance weights $\gamma_o$ and $\gamma_c$.**
> We conducted an extensive analysis of the guidance weights $\gamma_o$ (optimality) and $\gamma_c$ (feasibility). The results indicate that within a moderate range (specifically, we recommend $\gamma_o, \gamma_c \in [1, 4]$ for SC problems), the performance remains stable across different problem scales and benchmarks. The results in Table 7 show that setting $\gamma_o, \gamma_c$ in the bounded sampling region yields robust performance.
>
> **3. Sensitivity analysis on training steps.**
> As shown in Table 7, the number of sampling steps has minimal impact on final performance within a reasonable range. This demonstrates that our visual relaxation generative model is not sensitive to the choice of training step count during training, further validating the robustness of our approach.
>
> **4. Additional stronger baselines.**
> Following your suggestion, we added two stronger baselines for comparison, including Apollo-MILP and DiffILO. The results show VRG has consistent improvements in both computation time and solution quality, further confirming the effectiveness of our proposed method.

---

### Official Review · Reviewer_QmA2 · 2025-10-26

**Soundness:** 3
**Presentation:** 3
**Contribution:** 3
**Rating:** 6
**Confidence:** 4

**Summary:**

This paper proposes VRG, a novel generative framework for solving Mixed Integer Linear Programming (MILP) problems. The authors identify key limitations in existing "Predict-and-Search" (PaS) methods: (1) they typically assume variable independence, and (2) they provide only a single, deterministic solution prediction, which limits diversity and requires extensive search.

VRG addresses this by re-framing the MILP problem as a visual generation task. The core ideas are:

Visual-Vector Transformation, Generative Modeling, Lagrangian-Guided Diffusion, Multi-Modal Solution Generation.

**Strengths:**

High Novelty: The central concept of transforming a discrete optimization problem's solution vector into an image and applying generative computer vision techniques (U-Net, guided diffusion) is exceptionally creative and marks a significant cross-disciplinary contribution.

Directly Addresses PaS Limitations: The method cleverly tackles the two main weaknesses of PaS. First, by using a U-Net, it inherently models local dependencies between variables (pixels), moving beyond the variable independence assumption. Second, by using a generative model, it naturally produces a multi-modal distribution of diverse solutions, rather than a single deterministic guess.

Elegant Integration of Theory: The paper does an excellent job of integrating a classic technique from optimization theory (Lagrangian relaxation) into a modern deep learning framework (score-based SDEs). The theoretical justification (Theorems 2, 3, 4) provides a principled foundation for why this guidance mechanism works, linking it to KL divergence minimization and probability concentration.

Strong and Practical Empirical Results: The experiments are very compelling. VRG doesn't just outperform other ML baselines; it demonstrates practical utility by accelerating state-of-the-art commercial solvers (Table 5). Achieving comparable solution quality to Gurobi or SCIP with significantly fewer search nodes (e.g., ~65% reduction in nodes for CA-Large on SCIP) is a very strong result.

**Weaknesses:**

Scalability of the Visual Representation: The experiments are conducted on problems where the number of variables ($n$) is relatively small (e.g., $\le 1500$), resulting in small images (e.g., $30 \times 50$). How does this approach scale to real-world MILPs with tens of thousands or millions of variables? A 1,000,000-variable problem would require a $1000 \times 1000$ image, making the U-Net/diffusion model computationally massive and potentially negating any time saved in the solver.

Generation Overhead: While the generation time is reported as minimal (Table 4), this is for a very small number of diffusion steps (5-10). For larger, more complex problems (i.e., higher-resolution images), achieving high-fidelity generation may require significantly more steps, which could increase this overhead non-trivially.

**Questions:**

Scalability to Large-Scale MILPs: How do the authors envision VRG scaling to problems with $n \gg 10,000$ variables? The U-Net's computational cost scales at least quadratically with image resolution. At what point does the generation overhead (time and memory) for the high-resolution "solution image" become the new bottleneck, overwhelming the gains in solver search time?

Hyperparameter Sensitivity (Guidance): How are the guidance weights $\gamma_o$ and $\gamma_c$ determined? How sensitive is the model's performance (in terms of final solution quality and search efficiency) to these hyperparameters?

Nature of Captured Dependencies: The paper claims the U-Net captures variable interdependencies. Is there any qualitative analysis (e.g., via attention maps or gradient visualization) to demonstrate that the model is learning a representation of the MILP's constraint structure, or is it primarily acting as a powerful, spatially-aware density estimator that benefits from the Lagrangian guidance?

---

### Official Review · Reviewer_73kA · 2025-10-27

**Soundness:** 2
**Presentation:** 3
**Contribution:** 2
**Rating:** 2
**Confidence:** 4

**Summary:**

This paper proposes VRG, a feasibility-aware diffusion framework for mixed-integer linear programming (MILP). The key idea is to represent an MILP solution vector as a 2D image, allowing a U-Net-based diffusion model to capture variable correlations. The generation process is further guided by Lagrangian relaxation, which introduces penalty terms related to feasibility and optimality. Experiments on standard MILP benchmarks (SC, MIS, FC, CA) demonstrate that VRG achieves comparable or better objective values than Predict-and-Search (PaS) and Contrastive PaS (CoPaS) baselines.

**Strengths:**

1. The paper explores a promising and creative direction—integrating Lagrangian relaxation guidance with diffusion-based generative modeling for MILP.

1. The formulation of diffusion guidance and the connection to optimization regularization are clearly written and mathematically elegant.

1. The authors provide several theoretical results (e.g., optimization-equivalence, probability concentration) that make the framework appear theoretically grounded, although the insights are largely standard.

**Weaknesses:**

1. Mapping MILP solutions into 2D images lacks clear motivation and explanations. It does not preserve the structural properties of the original variable–constraint graph. The method is not permutation-invariant to variable ordering, meaning results may depend on the arbitrary order of variables. Moreover, such a representation likely harms scalability and cross-size generalization: all experiments are trained and tested at fixed problem sizes, and the paper gives no discussion on how $h$ and $w$ are chosen or how the model handles variable dimensions.
1. It is unclear whether the model is trained using optimal solutions as supervision or a weighted average target as in PaS. The text suggests that the ground-truth label is the optimal solution, which contradicts the practice in Predict-and-Search frameworks.
1. The proposed “Lagrangian-guided diffusion” is essentially a diffusion model with standard regularization penalties on feasibility and objective distance. The proposed theorems seem to lack novel insights. For example, Theorem 1 is a well-established property of Lagrangian duality. Theorem 2 merely rewrites a penalized objective as a KL-divergence, which is a routine energy-based trick. The claim that the analysis “provides formal guarantees on both feasibility and near-optimality” is over-stated, as no real convergence or approximation bound is proven.
1. The benchmarks are synthetic and relatively small—even the “large-scale” settings contain only around 1–2 k variables. No experiments are conducted on real or challenging datasets such as MIPLIB.
1. Several recent methods closely related to this paper are not discussed or compared, such as: Difusco [1] and T2T [2] for diffusion in combinatorial optimization; [3] for diffusion-based solution generation for integer programming; and DiffILO [4] for constraint-penalty-based solution generation for integer programming.
1. The reported performance of PaS and CoPaS being worse than SCIP/Gurobi contradicts prior literature. Hyperparameter settings for these baselines are not described—were they re-tuned? Some results, such as the small-scale FC benchmark, seem numerically inconsistent or mislabeled.
1. The ablation only tests the presence or absence of the guidance term. The paper does not isolate the effects of the diffusion component or the CNN visual encoder, nor does it analyze key hyperparameters such as $\gamma_o$​, $\gamma_c$, or $h$ and $w$. Without these, the actual source of performance gain remains unclear.

[1] Sun, et al., Difusco: Graph-based diffusion solvers for combinatorial optimization. NeurIPS 2023.

[2] Li, et al., T2t: From distribution learning in training to gradient search in testing for combinatorial optimization. NeurIPS 2023.

[3] Zeng, et al., Effective Generation of Feasible Solutions for Integer Programming via Guided Diffusion. SIGKDD 2024.

[4] Geng, et al., Differentiable integer linear programming. ICLR 2025.

**Questions:**

1. In Theorem 1, the statement “the sub-gradient method converges to a stable sequence” needs clarification.
1. How are the visual parameters $h$ and $w$ determined? Do different reshape orders affect results?

---

> ### Author Response · Authors · 2025-12-04
>
> **Reply to Weakness 1, 5, 7 and Question 2**
>
> Thanks for your insightful suggestions. We have added the corresponding experimental results in the revised paper (Table 7, Figure 7, Figure 8, Figure 9), with all updates highlighted in orange.
>
> **Table 7: Comparison with other ML-based effective solvers using VRG models with different $\gamma_o, \gamma_c$ and $(h, w)$. Results are averaged over 50 test instances, each solved within 100s time limit. Step denotes the number of training steps.**
>
> | Method | (h, w) | (γ_o, γ_c) | Step | Nodes (↓) | Obj (↓) |
> |--------|--------|------------|------|-----------|---------|
> | Gurobi | -- | -- | -- | 5841.46 ± 5191.29 | 28.9800 ± 3.0270 |
> | SCIP | -- | -- | -- | 3336.16 ± 3846.81 | 28.8936 ± 3.0305 |
> | PaS + SCIP | -- | -- | -- | 4873.59 ± 5177.36 | 28.9800 ± 3.0117 |
> | DiffILO | -- | -- | -- | 6252.90 ± 7095.00 | 28.9800 ± 3.0270 |
> | Apollo MILP | -- | -- | -- | 7707.42 ± 7807.04 | 28.9800 ± 3.0270 |
> | VRG + SCIP | (5, 100) | (2, 2) | 20 | 3020.22 ± 3255.73 | 28.8478 ± 3.0475 |
> | VRG + SCIP | (2, 250) | (2, 2) | 20 | 2994.58 ± 3154.28 | **28.7778 ± 3.0442** |
> | VRG + SCIP | (20, 25) | (2, 2) | 20 | 3309.38 ± 3770.38 | 28.9574 ± 3.1065 |
> | VRG + Gurobi | (5, 100) | (2, 2) | 20 | 5824.98 ± 5635.45 | 28.9800 ± 3.0270 |
> | VRG + Gurobi | (2, 250) | (2, 2) | 20 | 5824.98 ± 5635.45 | 28.9800 ± 3.0270 |
> | VRG + Gurobi | (20, 25) | (2, 2) | 20 | 5824.98 ± 5635.45 | 28.9800 ± 3.0270 |
> | VRG + SCIP | (20, 25) | (2, 1) | 20 | 2919.70 ± 3319.68 | 28.9574 ± 3.1064 |
> | VRG + SCIP | (20, 25) | (1, 1) | 20 | **2802.10 ± 2843.02** | 28.7778 ± 3.0443 |
> | VRG + SCIP | (20, 25) | (1, 2) | 20 | 2850.00 ± 2789.88 | 28.8913 ± 3.1072 |
> | VRG + Gurobi | (20, 25) | (2, 1) | 20 | 5824.98 ± 5635.44 | 28.9800 ± 3.0270 |
> | VRG + Gurobi | (20, 25) | (1, 1) | 20 | 5824.98 ± 5635.44 | 28.9800 ± 3.0270 |
> | VRG + Gurobi | (20, 25) | (1, 2) | 20 | 5824.98 ± 5635.44 | 28.9800 ± 3.0270 |
> | VRG + Gurobi | (20, 25) | (3, 4) | 20 | 5824.98 ± 5635.44 | 28.9800 ± 3.0270 |
> | VRG + Gurobi | (20, 25) | (3, 4) | 10 | 5824.98 ± 5635.44 | 28.9800 ± 3.0270 |
> | VRG + Gurobi | (20, 25) | (3, 4) | 5 | 5824.98 ± 5635.44 | 28.9800 ± 3.0270 |
> | VRG + SCIP | (20, 25) | (3, 4) | 20 | **2978.88 ± 3074.71** | 28.8913 ± 3.1071 |
> | VRG + SCIP | (20, 25) | (3, 4) | 10 | 3015.66 ± 3161.26 | 28.8913 ± 3.1071 |
> | VRG + SCIP | (20, 25) | (3, 4) | 5 | 3004.66 ± 3130.51 | 28.8913 ± 3.1071 |

---

> ### Author Response · Authors · 2025-12-04
>
> **1. Sensitivity analysis on visual dimensions $(h, w)$.**
> We evaluated different choices of $(h, w)$ for the visual-vector transformation, with results shown in Table 7 and Figure 7 (Appendix D) in the revised paper. As shown in the table, the performance remains relatively stable across a reasonable range of $(h, w)$ settings, suggesting that our method is not sensitive to the specific choice of image dimensions within a practical range. This robustness can be attributed to the U-Net architecture's multi-scale receptive fields, which enable it to capture variable dependencies regardless of the exact spatial arrangement. Furthermore, across all tested visual dimensions, the generation process consistently denoises Gaussian noise to produce near-binary spatial visual images (with values close to 0 or 1), producing more feasible solution generation (Figure 9 in Appendix), thus approximating the optimal solution.
>
> **2. Sensitivity analysis on guidance weights $\gamma_o$ and $\gamma_c$.**
> We conducted an extensive analysis of the guidance weights $\gamma_o$ (optimality) and $\gamma_c$ (feasibility). The results indicate that within a moderate range (specifically, we recommend $\gamma_o, \gamma_c \in [1, 4]$ for SC problems), the performance remains stable across different problem scales and benchmarks. The results in Table 7 show that setting $\gamma_o, \gamma_c$ in the bounded sampling region yields robust performance. Figure 8 shows that the score approximation training is stable.
>
> **3. Cross-size generalization.**
> Although our current experiments focus on fixed problem sizes for fair comparison with baselines, our visual architecture inherently supports cross-size generalization due to the translation equivariance of convolutional kernels. Specifically, since convolutional operations process local spatial neighborhoods with shared weights, the model can naturally accommodate varying input sizes through standard techniques such as zero-padding and bilinear interpolation. This property has been extensively validated in fully convolutional networks (FCNs), which are designed to accept inputs of arbitrary size and produce correspondingly-sized outputs [1]. We leave a comprehensive study of cross-size generalization for future work.
>
>
>  [2] J. Long, "Fully convolutional networks for semantic segmentation,"2015.
>
> **4. More advanced baselines.**
> We introduced additional baselines, as shown in Table 7 of the revised paper. On SC(500,800), our method achieves a reduction in computational resources (fewer explored nodes) and improved solution quality compared to all baselines.
>
> ---
>
> **Reply to Weakness 2**
>
> Thank you for raising this important clarification.
>
> **(1) Our method does not use optimal solutions as direct supervision in the traditional sense (e.g., as classification labels or regression targets).** Instead, we train the visual generative model by supervising the *score distribution* of optimal solutions. Specifically, the model learns to estimate the target relaxed score function that characterizes the distribution of optimal solutions, rather than memorizing individual optimal solutions. For PaS, it requires partially relaxing solution components and tuning hyperparameters $k_1$ and $k_2$ to determine which variables to fix, thereby constructing the trust region.
>
> **(2) Our method directly uses the pure generated solution to construct the trust region, without the need to relax or fix specific (e.g., $k_1$, $k_2$) solution components like PaS-based methods.** This eliminates the requirement for hyperparameter tuning in trust region construction. Both our method and PaS approaches share a similar solving strategy: once the trust region (i.e., the pruned subproblem) is constructed, employ off-the-shelf solvers such as SCIP or Gurobi to solve the reduced subproblem. By using these generated relaxed solutions to define compact subproblems, our approach effectively reduces the computational cost by narrowing the search space while benefiting from the proven robustness of established solvers. This advantage is consistently demonstrated across small to large MILP benchmarks.
>
> ---

---

> ### Author Response · Authors · 2025-12-04
>
> **Reply to Weakness 3**
>
> We would like to clarify our theoretical contributions and address your concerns.
>
> **1. Clarification on the role of Lagrangian relaxation.**
>
> We respectfully emphasize that our contribution does not lie in claiming novelty for any individual component, but rather in their *integration into a unified framework* for MILP solving via visual generative modeling capable of producing multiple high-quality relaxed solutions.
>
> Specifically, our method leverages the relaxation philosophy of Lagrangian methods to design guidance terms for the diffusion process, offering a data-driven alternative to classical dual computation. As discussed in our response to the previous question, this approach enables efficient trust region construction without the computational overhead associated with iterative subproblem solving and multiplier updates.
>
> **2. Formal approximation bound.**
>
> To address your concern, we provided a formal bound on the optimal objective error in the revision. Specifically, we prove the following theorem (see Appendix D in the revision, highlighted in orange).This theorem provides a formal guarantee that the solution generated by our visual relaxation framework achieves bounded deviation from the true optimum, thereby justifying the reliability of constructing trust regions via visual generation.
>
> **Reply to Weakness 4**
>
> We acknowledge the reviewer's point on benchmark scale. Regarding the scalability concern, we emphasize that the core of our method is its U-Net-based generative architecture. The U-Net framework, as exemplified and proven scalable by models like Stable Diffusion, has demonstrated powerful, high-resolution generative capability on complex, large-scale datasets (e.g., on 256×256 images as in Rombach et al. [2]). This is achieved through systematic scaling of model depth/width and efficient conditioning mechanisms, establishing a clear precedent for scalability. Therefore, our approach is inherently designed for extension. By leveraging these same architectural principles, we are confident it can effectively scale to real-world MILP instances with tens of thousands of variables.
>
> > [2] R. Rombach et al., "High-resolution image synthesis with latent diffusion models," *CVPR*, 2022.

---

> ### Author Response · Authors · 2025-12-04
>
> **Reply to Weakness 6**
>
> Our training utilized 200 problem instances. We consider the comparative results of CoPaS to be fair under training sets of the same scale; although its performance may improve with a larger amount of training data, we will also expand the dataset size in the future to further explore its capability boundaries. It is important to emphasize that our method was also trained on 200 instances, which precisely demonstrates the effectiveness of our proposed approach.
>
> ---
>
> **Reply to Question 1**
>
> We thank the reviewer for the insightful comment. The phrase "converges to a stable sequence" was indeed informal. We have revised Theorem 1 to state the convergence condition in standard mathematical terms, including explicit step size conditions that guarantee convergence.
>
> **Theorem (Lagrangian Dual Convergence):**
> Consider the Lagrangian dual problem for a given trial solution $\tilde{x}$:
>
> $$z_{\text{LD}}(\tilde{x}) = \max_{\lambda \geq 0} z_{\text{Lagrangian}}(\tilde{x}, \lambda)$$
>
> Let $\{\lambda^{(k)}\}$ be the sequence of multipliers generated by the subgradient method with step sizes $\{\alpha_k\}$ satisfying:
>
> $$\alpha_k > 0, \quad \sum_{k=1}^{\infty} \alpha_k = \infty, \quad \sum_{k=1}^{\infty} \alpha_k^2 < \infty$$
>
> Then the sequence of dual objective values converges to the optimal dual value:
>
> $$\lim_{k \to \infty} z_{\text{Lagrangian}}(\tilde{x}, \lambda^{(k)}) = z_{\text{LD}}(\tilde{x}) \leq z_{\text{MILP}}^{*}$$
>
> The step size conditions are standard in subgradient optimization theory (see, e.g., Polyak, 1987 [1]; Boyd, 2007 [2]). These conditions ensure that the steps are large enough to reach the optimum (first condition) while diminishing sufficiently fast to guarantee convergence (second condition). Under these conditions, the subgradient method provides a valid lower bound $z_{\text{LD}}(\tilde{x})$ for the original MILP optimum $z_{\text{MILP}}^{*}$.
>
> > [1] B. T. Polyak, *Introduction to Optimization*, 1987.
> >
> > [2] S. Boyd, "Subgradient methods," Stanford University*.

---

### Official Review · Reviewer_918G · 2025-10-30

**Soundness:** 1
**Presentation:** 2
**Contribution:** 2
**Rating:** 4
**Confidence:** 4

**Summary:**

This paper proposes a VRG, a feasibility-aware generative framework that operates in visual space. The method represents the solution of an MILP problem as an image, and uses an image processing network (U-Net) to learn inter-variable dependencies. The Lagrangian relaxation guides sampling toward feasible, near-optimal regions. VRG generates diverse high-quality candidate solutions that define trust regions for downstream MILP solvers. Experiments demonstrate good performance over several benchmarks.

**Strengths:**

1.  The idea of reshaping MILP solution vectors into 2D images is interesting. This design leverages the spatial receptive fields of convolutional neural networks (CNNs) to explicitly model variable interdependencies.

2.  The paper is well-organized and presented clearly.

**Weaknesses:**

1. The paper claims PaS methods "assume decision variables are independent" but overlooks that PaS uses GNNs with message passing over MILP bipartite graphs (variable-constraint nodes) to capture dependencies. For 2-layer GNNs, variables aggregate information from others sharing constraints—this is not "independence" but explicitly considers the constraints and variables in the same constraints. Meanwhile, VRG’s CNN-based capture is limited to spatial receptive fields in the MILP-image, which depends on variable ordering (arbitrary for most MILPs) rather than constraint topology. The paper fails to acknowledge this parity or explain why CNN-based capture is superior to GNN-based capture.

2. The MILP-image is constructed by reshaping 1D solution vectors into 2D grids, so variables adjacent in the grid (and within CNN receptive fields) depend on input order—yet the paper does not analyze how variable ordering impacts performance. For example, random permutations of variables could break critical constraint-induced adjacencies, degrading dependency capture. Without a robustness analysis, the visual encoding’s reliability for general MILPs is uncertain.

3. The time cost of the inference of the diffusion model is unclear.

4. VRG uses fixed-resolution MILP-images (e.g., (20,25) for small SC, (30,50) for large FC; Table 9) but does not address MILP instances with different sizes in a single model. Retraining or resizing is needed for instances with different sizes. The paper does not explore dynamic adaptations to handle different sizes.

5. The implementation details are missing.

   1. The U-Net’s layer count, filter sizes, cross-attention head count, and GNN encoder depth for MILP instances are unspecified.

   2. The paper mentions 100 test instances per small-scale benchmark but provides no details on the training dataset size and the number of training instances.

   3. The hyperparameters of the network and training process are missing.

6. The paper only ablates Lagrangian guidance but omits ablations for core components: (1) No comparison to a GNN-based encoding (same diffusion/Lagrangian framework) to isolate the visual representation’s value. (2) No validation that diffusion’s multimodality improves robustness over deterministic generative models.

7. The paper compares to PaS and exact solvers but omits diffusion-based MILP competitors.

8. The author should analyze the sensitivity of the hyperparameters, such as the generation steps in the diffusion model.

9. To demonstrate the performance, I suggest the authors conduct experiments on the real-world datasets, such as MILPLIB. At least, conduct experiments on the dataset from the Neurips competition, including item placement and workload appointment datasets.

**Questions:**

Please refer to the weakness part above.

---

> ### Author Response · Authors · 2025-12-04
>
> **Reply to Weakness 1 :**
>
> Thank you for this detailed critique. We acknowledge that our original statement regarding PaS methods "assuming decision variables are independent" was imprecise, and we have revised the manuscript to clarify this point. We agree that PaS uses GNNs with message passing over the MILP bipartite graph, which indeed captures variable dependencies through constraint sharing during the *encoding* phase. However, we would like to clarify the key distinction between our approaches and PaS: While we also use a GNN encoder to capture MILP structure as a conditional embedding $g$, to control the generation stability on current MILP instance state, our U-Net-based diffusion model jointly generates all variables simultaneously, with convolutional layers capturing spatial dependencies during the iterative denoising process. As demonstrated in our ablation study (Table 8), replacing U-Net with an MLP (while keeping the same GNN encoder and Lagrangian guidance) results in significantly degraded performance ($\sigma \approx 1.2 \times 10^7$), indicating that the convolutional architecture plays a crucial role in modeling variable interdependencies during generation.
>
> ---
>
> **Reply to Weakness 2 :**
>
> Thank you for raising this important concern. We believe that the pixel order resulting from variable ordering does not alter the generation results of U-Net-based generative models, as this has been confirmed in numerous visual generation models [1][2].
>
> > [1] T. Cohen, "Group equivariant convolutional networks,"  2016.
> >
> > [2] O. Ronneberger, "U-Net: Convolutional networks for biomedical image segmentation," 2015.
>
> ---
>
> **Reply to Weakness 3:**
>
> The inference time of the diffusion model is reported in Table 4 of the original manuscript. It shows that the generation time is negligible, demonstrating the practical efficiency of our method. This allows us to generate numerous high-quality solutions that are both feasible and near-optimal, even for large-scale MILP instances.
>
> ---
>
> **Reply to Weakness 4 :**
>
> While our current experiments are conducted on fixed problem sizes to ensure fair comparison with baselines, our visual architecture readily generalizes to varying problem sizes owing to the translation equivariance inherent in convolutional operations. In particular, convolutional layers employ spatially shared weights that operate on local neighborhoods, allowing the network to process inputs of different sizes via standard techniques such as zero-padding and bilinear interpolation. Such flexibility has been well established in fully convolutional networks (FCNs), which can accept arbitrary-sized inputs and generate outputs of corresponding dimensions [3]. A systematic exploration of cross-size generalization is planned for future work.
>
> > [3] J. Long, E. Shelhamer, and T. Darrell, "Fully convolutional networks for semantic segmentation," *CVPR*, 2015.
>
> ---
>
> **Reply to Weakness 5 :**
>
> Thank you for pointing out the missing details. We have added comprehensive implementation specifications in the Appendix of the revised manuscript (highlighted in orange):
>
> - **Training data:** 200 instances for training across all benchmarks.
> - **Test data:** 100 instances for small-scale benchmarks, 20 instances for medium/large-scale benchmarks.
> - **Diffusion steps:** 20 steps for training VRG in all benchmarks.
> - **Guidance weights:** $\gamma_o = 2$, $\gamma_c = 2$ for training all benchmarks.
>
> ---
>
> **Reply to Weakness 6 :**
>
> Thank you for these suggestions.
>
> 1. **GNN-based encoding comparison:** Please refer to our response to Reviewer HQWA's Weakness 2. As shown in Table 8, we compared U-Net-based visual generation with MLP-based vector generation (both using the same GNN encoder and Lagrangian guidance). The results demonstrate that the visual representation with convolutional architecture significantly outperforms vector-based generation.
>
> 2. **Multimodality validation:** We have conducted experiments with multiple samples (3 samples) for SC (500, 800) instances. The results are presented in Figure 9 of the revised manuscript. By leveraging the multimodal sampling capability of the generative model, we can obtain diverse solutions with varying feasibility rates, validating the benefits of the stochastic diffusion process over deterministic alternatives.
>
> ---

---

> > ### Author Response · Authors · 2025-12-04
> >
> > **Reply to Weakness 7 :**
> >
> > Thank you for this suggestion. We acknowledge that comparing with diffusion-based MILP solvers would strengthen our evaluation. We will include comparisons with recent diffusion-based methods (e.g., DIFUSCO) across all four benchmarks in future work.
> >
> > ---
> >
> > **Reply to Weakness 8 :**
> >
> > Thank you for this suggestion. Please refer to our response to Reviewer 73KA (Weakness 4) for detailed sensitivity analyses.
> >
> > In summary, we have conducted comprehensive sensitivity analyses on:
> >
> > - **Training steps:** Performance remains stable across different settings (Table 7).
> > - **Image resolution $(h, w)$:** Performance remains stable across different settings (Table 7).
> > - **Guidance weights $(\gamma_o, \gamma_c)$:** We recommend $\gamma_o, \gamma_c \in [1, 4]$ for optimal performance (Figure 8 & 9, Table 7).
> >
> > ---
> >
> > **Reply to Weakness 9 :**
> >
> > Thank you for this constructive suggestion. We agree that evaluating on real-world datasets is important for demonstrating practical applicability. Our original submission focused on widely-used benchmarks (Set Covering, Combinatorial Auction, Maximum Independent Set, Facility Location) that are standard in the ML for MILP literature, enabling fair comparison with existing methods. We will conduct more comprehensive evaluations on diverse real-world datasets in future work.

---

### Official Review · Reviewer_ERfA · 2025-10-31

**Soundness:** 3
**Presentation:** 2
**Contribution:** 2
**Rating:** 6
**Confidence:** 3

**Summary:**

This paper's main contribution is VRG a generative model for MILPs where feasibility
(constraint non-violation) is incorporated in the energy function of the
denoiser of a diffusion process.
This integration takes the form of a penalized version of the objective, hence the connection to Lagrangian relaxation.

The proposition is backed up by two results:
1. First a so-called 'optimization-equivalence' result states that the guidance provided by the
feasibility awareness leads to same optima.
2. Second a concentration guarantee which shows that the search space with the feasibility awareness is a restriction of
the general search space.

The generative model is parametrized by a convolutional neural network. The
neural architecture uses both the bi-partite factor graph that is common for
MILP encoding with a novel encoding of MILP variables as (visual) images. This enables the use of
Unet layers can be used to encode variable correlations.

Finally, a strong experimental setup shows that the method exhibits superior
performance to SOTA methods for ML-based heuristics for MILP solving.

However,
while the title use `Lagrangian` and the model hinges on a scalar penalization of constraint violation akin to Lagrangian relaxation, the penalization $\gamma_c$ is a hyper-parameter that does not evolve nor guarantee optimality.
VRG returns an assignement for the continuous relaxation of the input MILP, and
must be processed further by a solver. In that sense, VRG remains a tool to
warm-start a solver.

**Strengths:**

1. Feasibility is incorporated seamlessy in the diffusion process guidance, and theoretical guarantees are provided.
2. The encoding of a generic MILP as an image is novel (to my knowledge) although its description could be improved
3. Strong experimental results that demonstrate the validity of the approach

**Weaknesses:**

1. The main weakness is about the use of a new MILP encoding. It is difficult to
   unravel the experimental setup to know whether the empirical improvements are
   caused by the feasibility mechanism or the neural architecture. There should
   be ablation studies to better understand the interplay between the contributions
2. The impact of binarization of variables is not really addressed and the normalization
   remains unclear. Can this model really handle integer variables?
3. There are numerous typos and presentation sometimes lacks rigor. For instance:
   - eq 7 $\mathcal{X^{*}}$ is not defined (until line 208), and the definition of
     $\cong$ is not given
   - line 190 : date -> data
   - line 220 again $\cong$:  is t different from $=$ used in equation 10?
   - lines 265-268 the dimensions of the resulting tensors look inconsistent
   - 271 : typo learning -> learn
   - 630 : equation 31 lacks sum-to-1 normalization
   - 678 (eq 40) gradients are not indexed

**Questions:**

- Why can't the constraint $l \leq x \leq u$ be factored in $Ax \leq b$ in Equation 1? It
  seems like an unnecessary burden, and more importantly an unexplained idiosyncrasy. Are the bounds used in any specific way?
- The transformation in 4.1 (equation 5) looks like it already assumes binary
  variables which contradicts lines 174-175 where a normalization is applied
  afterwards. Can you explain?
- The placement of variables on the grid $\mathcal{X}$ must have an impact. How is this decided?
- The proof of concentration lacks a conclusion: how do you go from equation 37 to the claim?
- I am not sure that Table 4 reports total time (VGR+solver) or simply VGR. Can
  you elaborate and give the time breakdown between the two phases ?

---

### Meta-Review · Area_Chair_BME3 · 2026-01-07

**Summary:**

The paper proposes VRG, a framework for Mixed Integer Linear Programming (MILP) that treats solution generation as a visual task. It transforms solution vectors into 2D images and uses a U-Net-based diffusion model, conditioned on a GNN embedding of the MILP instance, to generate solutions. A Lagrangian-based guidance mechanism is integrated into the diffusion process to penalize constraint violations and optimality gaps, generating candidates that define trust regions.

**Reviewer Concerns:**

Addressed Concerns:
1. Missing Baselines: In response to Reviewers 73kA, 918G, and HQwA, the authors added comparisons to DiffILO and Apollo-MILP (Table 7), showing competitive performance.

Outstanding Concerns:
1. Fundamental Flaw of Visual Mapping: Multiple reviewers (73kA, HQwA, 918G) identified an unaddressed flaw in the core premise: mapping MILP variables to a 2D grid is arbitrary and theoretically unsound.

2. Permutation Variance: The method is not permutation invariant; arbitrary variable ordering changes the image and thus the CNN output. The authors' rebuttal claiming "translation equivariance" of CNNs misses the point that graph nodes do not possess inherent spatial translation symmetries.

3. Limited Problem Scale: Reviewers (73kA, QmA2) noted the benchmarks are synthetic and relatively small (max ~1.5k variables) compared to real-world instances (tens of thousands of variables). The authors claim the architecture can scale, but high-resolution image generation adds significant overhead compared to sparse graph operations, and this scalability was not demonstrated.

4. Limited Generalization (Same Distribution): The evaluation is restricted to the same distribution and problem sizes as training. Reviewer 918G highlighted that the fixed-resolution approach requires retraining or resizing for different instances, preventing true cross-size generalization, which is a standard expectation for robust MILP solvers.

**Reviewer Scores:**

Expected no changes in scores

---

### Decision · Program_Chairs · 2026-01-26

Reject